# Dynamic Window-level Granger Causality of Multi-channel Time Series

## Abstract

Granger causality (GC) techniques facilitate the examination of temporal causal relationships by assessing the predictive utility of one time series for another. However, conventional GC approaches are limited to linear scenarios and assume that causalities exclusively exist between entire time series channels, remaining constant over time. This assumption falls short when dealing with real-world time series data characterized by dynamic causalities among channels. To address this limitation, we present the dynamic window-level Granger causality with causality index (DWGC-CI) method, which incorporates nonlinear window-level variability into the traditional GC framework. The DWGC-CI method specifically employs F-tests on sliding windows to assess forecasting errors, subsequently extracting dynamic causalities using a causality index. This index involves the introduction of a negative feedback adjustment mechanism, which mitigates window-level noisy fluctuations and helps extract dynamic causalities. We theoretically demonstrate that, compared to traditional channel-level GC methods, DWGC-CI accurately captures window-level GCs. In experimental evaluations involving two synthetic datasets, DWGC-CI significantly surpasses the baseline in terms of accuracy and recall rates. Furthermore, when applied to two real-world datasets, DWGC-CI effectively identifies seasonal and annual varying GCs, demonstrating a markedly better consistency with domain-specific knowledge of seasonal meteorology and stock market dynamics.

## 1 Introduction

Time series data, characterized by a pre-defined temporal or sequential order Hamilton (1994), is extensively employed in a diverse range of real-world applications, encompassing signal processing Scharf (1991), economics Granger & Newbold (2014), and climatology Mudelsee (2013), among others. Common tasks involving time series data consist of indexing, clustering, classification, regression Keogh & Kasetty (2003), and causality detection Eichler (2012). Among these tasks, causality detection, which emphasizes cognitive reasoning Pearl (2018), offers a more comprehensive understanding of system behaviour compared to traditional statistical machine learning and deep learning methods that rely on correlation fitting. Owing to the inherent nature of temporal precedence in time series data, it not only encapsulates empirical patterns of variable changes but also captures prior knowledge of causal relationships among distinct channels Eichler (2013).

To deal with the causal reasoning task in time series, Granger Granger (1969) first proposed the statistical test to decide whether one time series channel is influential when predicting another, which is known as Granger causality (GC). With the GC method, the causality can be obtained from the temporal data in various domains, such as physics Kaufmann & Stern (1997), biology Angelini et al. (2009) and economy Comincioli (1996).

Nonetheless, conventional methods Xu et al. (2016); Wu et al. (2022); Granger (1969) presuppose that GC remains invariant across time series channels over time. It is a concept referred to as "channel-level GC", more precisely. While this assumption facilitates the resolution of numerous Granger tests, this channel-level static assumption is not obligatory. It mainly serves merely to streamline the Granger testing process Granger

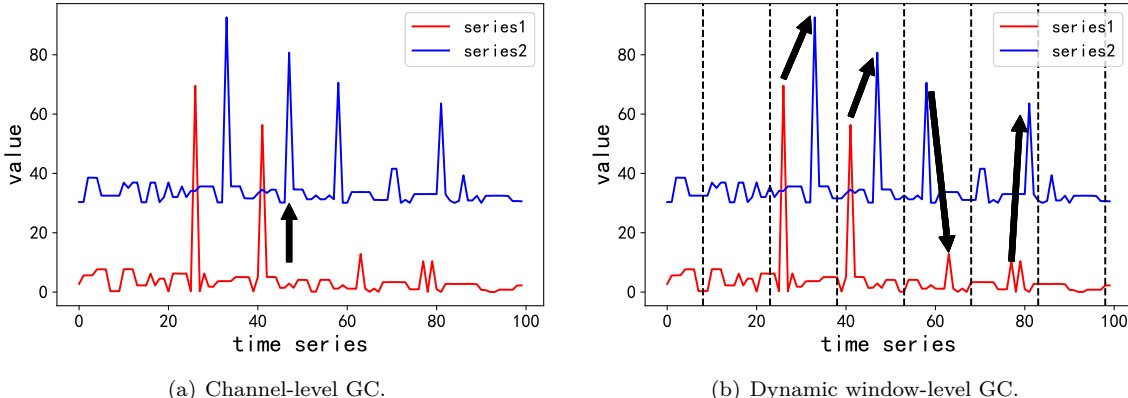

(a) Channel-level GC.  (b) Dynamic window-level GC.

Figure 1: The channel-level approach (a) assumes that causal influences between time series channels remain consistent throughout the duration of the series. In contrast, the window-level causality method (b) involves scrutinizing the dynamic evolution of causal interactions within sliding windows applied to the time series data. This latter approach allows for a more nuanced understanding of shifting causality patterns (shown as four-wave crests) over time.

(1980). More crucially, this assumption contradicts numerous practical situations involving dynamic causalities. In reality, time series data has grown increasingly voluminous, intricate, and uncertain, causing causality relationships to evolve in conjunction with the sliding windows of time series data (refer to the four-wave crests as in Fig. 1). Notable applications encompass seasonal meteorology and stock market dynamics, which will be detailed in Sec. 5. Consequently, dynamic window-level causalities, when offering reliable and precise predictions, are better suited for real-world applications.

Addressing the challenge of time-varying window-level causalities, preliminary efforts have been made on the specific application domains. One notable example is the field of neuroscience, where the "dynamic causal modelling" (DCM) method Friston et al. (2003) identifies dynamic causal relationships between neuronal clusters by formulating differential equations that capture neural dynamics. However, the DCM approach is grounded in a domain-specific hypothesis-testing framework, rendering it unsuitable for the broader GC context. Stepping backwards, although alternative time-varying GC methods have been proposed Cekic et al. (2018); Pagnotta et al. (2018); Ming-Hsien & Chih-She (2015), these approaches predominantly rely on linear forecasting models and inadequately account for local fluctuations. For example, it is challenging to linearly model the series data depicted in Fig. 1, and more importantly, such fluctuations might result from random noise rather than causal relationships, leading to significant inaccuracies in local-window GC. In summary, current research on the theoretical underpinnings and extraction techniques for window-level time-varying GC is not mature enough, necessitating further investigation.

## 1.1 Our proposal

In this study, we propose a novel approach to GC analysis by introducing dynamic window-level GC with causality index (DWGC-CI), which extends the conventional GC framework by relaxing the assumption of constant causality on sufficiently long time series Granger (1980). Our method is insensitive to the selection of the prediction model, and we adopt the nonlinear autoregressive (NAR) model Chen et al. (2000) for simplification. Subsequently, we first present the dynamic window-level GC (DWGC) method, which employs the window-level F-test to compare prediction results on sliding windows, with GC being a special case when the entire channel is treated as a single window. Second, to mitigate potential fluctuations in the window-level F-test, we further establish a causality index matrix by optimizing its associated index loss, culminating in the complete DWGC-CI methodology.

In this study, we will provide a comprehensive elucidation of the theoretical guarantees and practical advantages of DWGC-CI. Theoretically, we demonstrate that: 1) the recall rate of DWGC increases with window length, ultimately converging to that of GC; It mathematically reveals a *granularity-efficacy trade-off*, which

drives the proposal of the DWCI-CI technique. By incorporating supplementary CI, our DWGC aspires to overcome this trade-off and 2) surpasses DWGC/GC in terms of recall rate. Experimentally, to evaluate the performance of DWGC-CI, we conduct experiments on four datasets, including two synthetic and two real-world datasets. For the synthetic data, we employ linear autoregressive (AR) and NAR simulators for generation, with DWGC-CI achieving superior window-level causality detection compared to the baseline, regarding the recall rate and accuracy rate. In the case of real-world datasets, DWGC-CI effectively uncovers dynamic causalities in the El-Niño and East Asian monsoon phenomena, as well as the relationship between the stock market and the economy. These findings are consistent with expert knowledge previously reported in the literature. In summary, the contributions of this paper are threefold:

- We are the first to propose a general dynamic window-level GC detection method taking advantage of the causality index, called DWGC-CI in total.

- We provide the theoretical proof demonstrating that the DWGC method contains a granularity-efficacy trade-off. Moreover, the recall rate of DWGC can be enhanced via utilizing the causality index trick, which is integrated within the final comprehensive DWGC-CI methodology.

- We evaluate the performance of our DWGC-CI method through a series of numerical experiments involving two synthetic and two real-world datasets. Our findings reveal that the DWGC-CI approach successfully identifies window-level causalities that align with previously established academic knowledge.

## 2 Preliminaries of Granger Causality

We use $(Y_{i,1}, Y_{i,2}, \cdots, Y_{i,t}, \cdots)$ to denote the time series channel $i \in \{1, 2, ...d\}$, where $Y_{i,t}$ and $Y_{i,<t}$ are the series channel $i$ at and before time $t$. [1] In this paper, we use the double-headed arrow to represent the "channel-level GC" Granger (1969) from channel $j$ to $i$: $Y_j \Longrightarrow Y_i$. The GC from the series channel $j$ to $i$ holds if channel $j$ provides useful information when predicting the future values of series $i$. In detail, the traditional GC uses an AR model $g$ to predict future values:

$$\hat{Y}_{i,t} = g_i(Y_{i,<t}), \hat{Y}_{i|j,t} = g_i(Y_{i,<t}, Y_{j,<t}). \tag{1}$$

Then GC can be built based on an F-test to check the difference between $\hat{Y}_{i|j,t}$ and $\hat{Y}_{i,t}$, i.e., the forecasting result of the channel $i$ with and without channel $j$:

$$F_{j \Longrightarrow i} = \sum_t (\hat{Y}_{i,t} - Y_{i,t})^2 / \sum_t (\hat{Y}_{i|j,t} - Y_{i,t})^2, \tag{2}$$

where $g_i$ is the prediction on time series $i$ (e.g., AR model in traditional GC method). The criteria of this hypothesis test is that we accept that GC $Y_j \Longrightarrow Y_i$ exists when $F_{j \Longrightarrow i}$ in 2 is larger than a pre-defined causality threshold $\epsilon$ ($e.g., \epsilon \geq 1$).[2]

The generalization of GC methods can be categorized into two primary domains: window-level and nonlinear. However, achieving a concise and effective integration of both within the general GC framework remains a challenging task. Regarding window-level GCs, the hypothesis-testing framework of dynamical causal modelling (DCM)Friston et al. (2003) is incompatible with the general GC framework. Additionally, other time-varying methodsCekic et al. (2018); Pagnotta et al. (2018); Ming-Hsien & Chih-She (2015) simply incorporate time-varying coefficients in regression models, but they exhibit suboptimal performance in handling irregular nonlinear cases. As for nonlinear GCs, Chen et al.Chen et al. (2004) employed a delay embedding technique to extend nonlinear Granger causality. Later, Sun et al.Sun (2008) suggested using

---

[1] In practice, we follow the GC's basic assumption–the time series $Y$ is stationary; for the generating process, the joint probability distribution function of the piece of time series does not change over time. In order to achieve this objective optimally, it is imperative that we conduct stationery tests previously and subsequently perform appropriate transformations to ensure stationery.

[2] The well-known premise of 2 is that series channels $i$ and $j$ meet the "backdoor condition" Pearl & Robins (1995), shown as Appendix. G.

kernel embeddings to address the nonlinearity. Subsequently, Sugihara et al.Sugihara et al. (2012) expanded the traditional GC to the Convergent Cross Mapping (CCM) theorem, which is based on Taken's embedding theoremTakens (1981). Moreover, there are additional attempts via tools such as transfer entropy Syczewska & Struzik (2015), causal structure learning Weichwald et al. (2020), spectrum-based framework Zhang et al. (2016). Eichler Eichler (2013) subsequently re-summarized the challenges associated with transferring GC to the nonlinear version, which includes: 1) aggregating the time-varying coefficients required by Granger causality tests and 2) the instability of the causal structure itself. More recently, neural networks have been introduced Tank et al. (2018); Xu et al. (2019); Duggento et al. (2019); He et al. (2019) to supplant the original autoregressive (AR) model for sequence fitting and prediction. Regrettably, the combination of window-level GC and nonlinear GC is quite challenging since it is hard to get through the local noisy information of complex data to extract informative GC. To our best knowledge, no absolutely universal and simple framework has yet been developed yet, and currently, there are only several attempts in specific engineering applications (such as EEG signals Li et al. (2012), switching systems Ma et al. (2014)).

In addition to GC, causal graph modelling serves as an alternative approach for discerning causal relationships between distinct time series channels Xu et al. (2019). Causal graph models can be constructed using various methodologies, including vector autoregressive Granger analysis (VAR), generalized additive models (GAMs) Lütkepohl (2005), and specific predefined regression models Sommerlade et al. (2012). However, generating a causality graph for each time point presents a significant challenge due to the increased computational complexity, particularly when identifying dynamic window-level causality. Furthermore, after graph construction, A. Mastakouri Mastakouri et al. (2021) theoretically analyze the condition of dynamic causality identification in a time series graph, whereas their theorem is based on complex assumptions which are computationally hard to verify in the real world (such as considerable conditional independence tests), and it is also challenging to concurrently obtain all window-level causal relationships.

## 3 Method

### 3.1 DWGC model

To detect dynamic window-level causality between two data series $Y_i$ and $Y_j$, we first define sliding windows of length $k$ on the same time position $t$ of both series:

$$\{Y_{i,t}, Y_{i,t+1}, \cdots, Y_{i,t+k-1}\}, \quad \{Y_{j,t}, Y_{j,t+1}, \cdots, Y_{j,t+k-1}\}.$$

Then we consider two forms of time-series fitting on the sliding windows on channel $i$ with or without the information from channel $j$. Specifically, to fit the nonlinearity of the data, we use the NAR model as $g$ in 1, and modify 2 as follows:

$$F_{j \Longrightarrow i}^{t,k} = \sum_q (\hat{Y}_{i,q} - Y_{i,q})^2 / \sum_q (\hat{Y}_{i|j,q} - Y_{i,q})^2, q \in [t, t+k-1], \tag{3}$$

where $\hat{Y}_{i,q}$ and $\hat{Y}_{i|j,q}$ denote the predictions on the local sliding window with and without the information of channel $j$, respectively. $F_{j \Longrightarrow i}^{t,k}$ denotes the F-statistic on the sliding window $[t, t+k-1]$ from channel $j$ to $i$. Analogously to 2, the criteria is that we accept the window-level causality $Y_{j,t \sim t+k-1} \Longrightarrow Y_{i,t \sim t+k-1}$ exists if $F_{j \Longrightarrow i}^{t,k}$ is larger than the pre-defined threshold $\epsilon$ ($\geq 1$). In this sense, GC is the special case of the DWGC method when $k$ is equal to the channel length.

Noteworthy, DWGC makes a compromise on time-invariance and varying and takes advantage of the information of the channel up to not only the information of the current window but also the information of the channel up to time point. The task of DWGC is to accurately determine whether channel has received a causal effect within the window $t \sim t+k-1$.

### 3.2 DWGC-CI model

In the preceding section, it was observed that the reduction in length from the channel (Eqn.2) to window (Eqn.3) renders the DWGC method susceptible to instability arising from NAR forecasting results.

To address this issue, we introduce the "causality index (CI) matrix", designed to mitigate the temporal fluctuations of NAR forecasting outcomes. This approach can be likened to a negative feedback regulation mechanism, wherein iterative training is performed, and varying weights are assigned to each time point to stabilize the overall F-statistic results.

Specifically, each original point $Y_{i,t}$ could be weighted by an index matrix denoted as $\Phi = \{\Phi_{i,t}\}$, which is taken as an additional input layer of the NAR model, the weighted item $\{\Phi_{i,t} * Y_{i,t}\}$ is denoted as:

$$Y_{i,t\sim t+k-1}^{\Phi} = \Phi_{i,t\sim t+k-1} * Y_{i,t\sim t+k-1}, \ i = 1, 2, ...d, \tag{4}$$

where $Y_{i,t\sim t+k-1}, \Phi_{i,t\sim t+k-1}, Y_{i,t\sim t+k-1}^{\Phi}$ are the column vectors consisting of $k$ elements in the window $[t, t + k - 1]$, denoting original data, weighting factor and weighted data respectively, and $*$ is the Hadamard product. We establish a minimization goal to learn the indexing matrix $\Phi$ to scale down the temporal fluctuations:

$$L_{index} = \sum_{i,q \in M} \left\| \Phi_{i,q} - h \left( \left| (\hat{Y}_{i,q}^{\Phi} - Y_{i,q}^{\Phi})^2 - \mathbb{E}_q(\hat{Y}_{i,q}^{\Phi} - Y_{i,q}^{\Phi})^2 \right| \right) \right\|_2^2, \tag{5}$$

where $M$ is the set of sliding windows which detected causality and $h$ is a monotonically decreasing function[3]. On this basis, the effectiveness of the "causality index (CI) matrix" can be explained as a *dynamic negative feedback regulation*: In each iteration, it aims to regulate overall forecasting errors on each window which detected causality by reducing the effect of adverse factors such as local noise and local over-fitting. For instance, when the term $(\hat{Y}_{i,q} - Y_{i,q})^2$ is much larger (noise) or smaller (over-fitting) than others, which is a statistically significant anomaly, then $\left| (\hat{Y}_{i,q} - Y_{i,q})^2 - \mathbb{E}_q(\hat{Y}_{i,q} - Y_{i,q})^2 \right|$ will be significantly large. We use the negatively correlated index $\Phi_{i,q}$ to superimpose it to weaken its fluctuation on the local-window F-statistics.

**Remark 1.** *We provide the further justification of $L_{index}$.*
*A: The reason for not updating CI on the entire area. Noteworthy, we aim to increase the credibility of areas where there may be potential causal relationships with significant fluctuations by excluding the influence of noise. The specific approach is to weaken the impact of excessive outliers in these "suspicious areas" (as they are likely noise). This compels us to focus on updating the coefficients of the CI matrix in the "suspected" regions of causal relationships rather than considering the entire area. It should be noted that this weighted filtering pattern is dynamic because the extracted set of causal relationship windows may vary in each round. Our experiment results could further demonstrate this motivation since the performance is weaker if we conduct an update for all windows.*
*B: The reason for comparison with $\mathbb{E}_q(\hat{Y}_{i,q}^{\Phi} - Y_{i,q}^{\Phi})^2$: Due to the unstable and easily disrupted nature of DWGC under window settings, when this indicator significantly increases, there are two factors at play: a) the genuine causal effect (observing the influences), and b) the local window noise (observing outliers of the instantaneous errors in the auto-regressive model). The calculation of differences between instantaneous errors and the expectation over instantaneous errors aims to disentangle these two factors. Specifically, in the windows where potential causal relationships exist, we actively detect significant outliers in this set of instantaneous errors. These outliers are more likely to be influenced by real-time noise compared to other points. Through a negatively correlated (monotonically decreasing) function h, we apply negative feedback adjustment to the weighted values (CI matrix) corresponding to these outliers. This aids in reducing the impact of these noisy points on the final result's weight (since the absolute prediction error is often proportional to the scale of the data points themselves).*

On this basis, the window-level F-statistics in each iteration is addressed by the F-test method analogously to 3, and we denote the training data/model outcome in each iteration as $Y_{i,q}^{\Phi}, \hat{Y}_{i,q}^{\Phi}, \hat{Y}_{i|j,q}^{\Phi}$. We adopt it to iteratively determine the window set $M$ in 5:

$$F_{j \Longrightarrow i}^{t,k} = \sum_q (\hat{Y}_{i,q}^{\Phi} - Y_{i,q}^{\Phi})^2 / \sum_q (\hat{Y}_{i|j,q}^{\Phi} - Y_{i,q}^{\Phi})^2, q \in [t, t+k-1]. \tag{6}$$

---

[3]The choice of $h$ is not unique and in this paper we set $h(\cdot) = (\alpha - \tanh(\cdot)), \alpha > 1$.

---

**Algorithm 1** DWGC-CI method

---

**Data:** Multi-channel time series $\{Y_{i,t}\}$ and predefined F-test threshold $\epsilon$.

Initialization, $\Phi$ set to all-one;

  **while** *NAR forecast loss and $L_{index}$ converge* **do**

    |   Reweight the original time series using causality index matrix $\Phi$ via 4;

    |   Train an NAR model using the reweighted series $\{Y_{i,t}^{\Phi}\}$;

    |   Finding dynamic causalities via 6;

    |   In all the window pairs with detected causalities, optimize $L_{index}$ in 5 to update $\Phi$;

**end**

**Result:** Window-level GC $\{Y_{j,t \sim t+k-1} \Longrightarrow Y_{i,t \sim t+k-1}\}, i, j \in \{1, 2, ...d\}$.

---

In summary, the DWGC-CI method uses NAR's prediction errors to reversely control the causality index, which can be seen as the input layer weights of the NAR model, and detect causalities in this process. The final procedure for the conclusion is outlined in Algorithm 1. [4]

### 3.3 Practical Issues

**Selection of sliding window size** The optimal selection of the sliding window size can be best decided based on the background knowledge of the GC period. In the following experimental part, for a monsoon investigation, we employ a monthly sliding window to investigate the causality of seasonal fluctuations, whereas, in a stock market analysis, an annual sliding window is utilized to examine the causality of yearly variations. We will illustrate in Appendix. E.1 that, when our choice of sliding window size is clearly not in line with a priori intuition, it is likely that we will not reach an informative conclusion about GC.

In cases where background knowledge is not accessible, our approach relies on previously established sliding window size selection methods found in the time series forecasting literature, which could potentially yield suboptimal outcomes. These methods encompass: 1) preemptively categorizing point-in-time sample features to ensure that samples sharing similar attributes are grouped within the same window; 2) introducing minor perturbations to the dataset and evaluating the robustness of DWGC-CI, with the optimal window length corresponding to the most robust result; and 3) implementing a ROC-based technique to determine the appropriate sliding window size, as detailed in reference Fkih et al. (2012).

**Precision-recall curve** The precision and recall rates are both intimately related to the selection of the specific threshold value, denoted as $\epsilon$. As $\epsilon$ increases, and the likelihood of a window being accepted as window-level GC diminishes, leading to a decrease in the recall rate. Simultaneously, due to the conservative nature of the decision criteria, the precision rate is expected to increase correspondingly. A thorough precision-recall trade-off with $\epsilon$ will be presented in Appendix. E.2, in which we also show that DWGC-CI contains a relatively low false-positive rate without GC ground truth between two series.

## 4 Theoretical Analysis

In this section, we give the theoretical analysis of our methods. First, we present an important and interesting 'Negative Result' regarding the performance of traditional GC methods at the window-level: their performance gradually worsens as the window size decreases, which aligns with our intuition that "window-level instability" is prone to occur; 2) This negative result serves as a solid foundation for the innovative design of DWGC-CI. We need to introduce a new force, namely the dynamically adjustable CI matrix, to counteract the diminishing power of the GC model as the window size decreases.

Intuitively, the robustness and stability of DWGC are in direct proportion to the window length and can be enhanced by the causal index (CI) based on the view of negative feedback regulation. Mathematically, we re-organized the statement into two theorems: 1) DWGC's recall rate increases with window length

---

[4]In addition, compared with the winsorization Dixon & Yuen (1974) method, our causality index can effectively capture the causal information of each window, while winsorization only concentrates on several points with abnormal prediction errors in each iteration.

and degenerates to GC's, and 2) DWGC-CI's recall rate surpasses DWGC's in window length and GC's in channel length.

For definitions, we analytically consider the series channel length as $L$ and the sample pairs $T_j, T_i$. Naturally, the recall rate as below refers to the overall probability of detecting $F_{j \Longrightarrow i} > \epsilon$ on retrieved windows with causalities. The definition of accuracy is analogous.

### 4.1 DWGC: granularity-efficacy trade-off

We first analyze the granularity-efficacy trade-off of DWGC. We choose the sliding window length as granularity and the recall rate as efficacy. We refer readers to some additional criteria of efficacy (such as accuracy rate) in the experiment part.

**Assumption 1.** *The output of the machine learning model $g(\cdot)$ approximates a Gaussian distribution. Namely, $g_i(Y_{i,<t}) \sim \mathcal{N}(Y_{i,t}, \cdot)$, $g_i(Y_{i,<t}, Y_{j,<t}) \sim \mathcal{N}(Y_{i,t}, \cdot)$.*

**Theorem 1** (Positive correlation between recall rate and window length of DWGC)**.** *Suppose Assumption. 1 holds. Assume the channel level GC exists (we have enough confidence to reject it only when $F_{j \Longrightarrow i} < \epsilon$); moreover, the outputs satisfy an unbiased and stable Gaussian distribution with the same magnitude of fluctuations at each data point, i.e., the Gaussian distributions of these outputs share a uniform noise variance when conducting a same regression model. Then the DWGC's recall rate satisfies $\mathbb{P}(F_{j \Longrightarrow i}^{t,k_2} > \epsilon) - \mathbb{P}(F_{j \Longrightarrow i}^{t,k_1} > \epsilon) = \mathcal{O}(\frac{k_2 - k_1}{2^{k_1}})$. Specifically, $\forall k_1, k_2 \in [0, L], k_1 < k_2$,*

$$\mathbb{P}(F_{j \Longrightarrow i}^{t,k_2} > \epsilon) - \mathbb{P}(F_{j \Longrightarrow i}^{t,k_1} > \epsilon) \in \left[ 0, C(k_2 - k_1) 2^{-k_1} \right], \text{where } C \text{ is a constant.} \tag{7}$$

**Corollary 1** (Granularity-efficacy trade-off)**.** *DWGC's recall rate monotonically increases with (efficacy increases) the sliding window length $k$ (granularity decreases). Naturally, DWGC degenerates to the traditional GC method in the process of window length $k$ approaching $L$.*

The strict proof is referred to in Appendix B-D. This theorem encapsulates a critical insight: directly transitioning the GC detection from channel-level to window-level may imply a significant compromise in performance, particularly with regard to the recall rate. Owing to the "*trade-off*" between the granularity of the results and the efficacy, we are compelled to propose the DWCI-CI method. We endeavour to strike this "*trade-off*" by introducing additional CI, thereby controlling local noise. We assert that, through the appropriate selection of CI, the performance of DWGC-CI will surpass that of DWGC, which will be illustrated in the following part. We also refer readers to Fig. 2 for deeper comprehension.

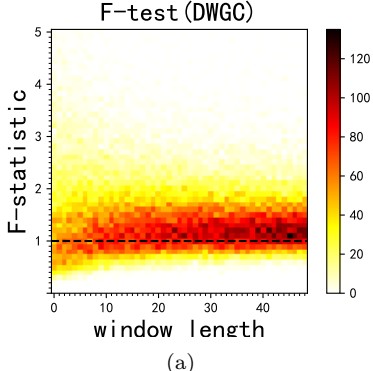
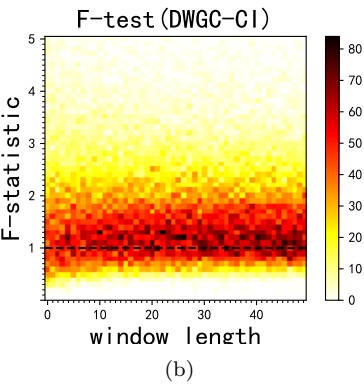

Figure 2: (a) shows the granularity-efficacy trade-off of DWGC on a simulation dataset (NAR dataset generation in Sec 5.1). The recall rate $P(F_{j \Longrightarrow i}^{t,k} > \epsilon = 1)$ is monotonically increasing with the window length $k$, and when k is large (e.g., $k > 20$), its performance stabilises quickly. This is consistent with the monotonicity and convergence property as reflected in Corollary. 1. Based on this insight, DWGC-CI (b) is exactly what helps to improve the performance of DWGC, especially in smaller windows (e.g., $k < 20$).

### 4.2 DWGC-CI: enhancing the granularity while keeping efficacy

In the above section, we introduce the granularity-efficacy trade-off implicated by DWGC. In this part, we claim the causality index (CI) can help enhance the efficacy (recall rate) while keeping the granularity (window level).

**Theorem 2** (DWGC-CI outperforms DWGC)**.** *Suppose Assumption. 1 holds. Assume the window length is chosen as $k$, and the dynamic window-level GC on $[t, t+k-1]$ exists (we only have enough confidence to reject it only when $F_{j \Longrightarrow i}^{t,k} < \epsilon$); moreover, if the data are weighted using the CI-matrix, the output will also be a Gaussian distribution with the same scale of deflation as the corresponding CI-matrix. Then, the causality index (CI) matrix $\Phi$ exists to make DWGC-CI's recall rate outperform DWGC's.*

We prove it via constructing sufficient conditions and leave the proof in Appendix C. Intuitively, we impose smaller CI to the time point with local fluctuations (noise/overfit) in order to make stabilization, and hence the F-test is more informative and the recall rate is enhanced. Seen from Fig. 2, most F-statistics of DWGC-CI are steadily higher than DWGC's on the same window length $k$, especially when $k$ is relatively small (e.g., $k < 20$). Moreover, in the process of training approximate $\Phi$, we adopt the sufficient condition of $\Phi$ as the regularization items for 5 (see Appendix. C for detail) and conduct automatic differentiation.

Honestly, it is acknowledged that there still lacks rigorous convergence conditions of CI matrix due to the lack of clear mathematical tools, which presents an promising theoretical direction for future work. However, it is gratifying that in experiments, DWGC-CI has demonstrated superior performance, confirming that good CI is often easy to obtain.

## 5 Experiments

In this section, we employ our proposed techniques to analyze two synthetic and two real-world datasets. As previously elucidated in the preliminary discussion, the integration of window-level and non-linear approaches represent a novel avenue of research, which consequently leads to a scarcity of well-established baselines. To ensure rigorous and impartial evaluation, our experimental analysis primarily concentrates on validating the theoretical propositions (Sec. 5.1) and assessing the congruence with a priori knowledge (Sec. 5.2).

### 5.1 Experiment on synthetic AR/NAR simulation dataset

We apply DWGC/DWGC-CI on the synthetic (N)AR dataset. These results further verify the theoretical results in the last section.

#### 5.1.1 Experimental setting

The AR and NAR simulations generate two synthetic datasets, and the generation process is detailed as follows.

**1)** *The linear AR simulation construction*: We simulate two linear AR time series with a lag value randomly picked from the integers from one to nine ($t = 0, 1, ...$): $T_{1,t} = m_1 T_{2,t-lag} + 0.02 n_1(t)$, $T_{2,t} = m_2 T_{1,t-lag} + 0.02 n_2(t)$. For $i = 1, 2$, $n_i(t)$ is the Gaussian noise term ($n_i(t) \sim N(0,1)$), and the initial values are $T_{i,t} = t \mathbb{1}_{t \in [1,5]} + (11-t) \mathbb{1}_{t \in [6,10]}, m_i = 0.9 \mathbb{1}_{rand(0,1) \le 0.95} + 10 \mathbb{1}_{rand(0,1) > 0.95}$.[5] **2)** *The NAR simulation construction*: we simulate two NAR time series: $T_{1,t} = \text{Re}(m_1 \sqrt{T_{2,t-lag}^2 - 1} + n_1(t))$, $T_{2,t} = \text{Re}(m_2 \sqrt{T_{1,t-lag}^2 - 1} + n_2(t))$, where $n_1(t)$, $n_2(t)$, the lag and the initial value of $T_1, T_2$ are the same as AR's. Let $f(s) = \sin(0.1s)$, $s$ is random. Then the values of $m_1, m_2$ are $m_1 = 10 \mathbb{1}_{f(s) > 0.9} + 0.9 \mathbb{1}_{f(s) < 0.9}$, $m_2 = 10 \mathbb{1}_{f(s) < -0.9} + 0.9 \mathbb{1}_{f(s) > -0.9}$. The time point of $m_i = 10, i = 1, 2$ denotes the beginning of the causality[6]. At last, $\text{Re}(\cdot)$ takes the real part of the square root.

---

[5]Let $\mathcal{X}$ be the set of possible values of $x$, then $\mathbb{1}_{x \in \mathcal{X}}$ denotes the characteristic function. Moreover, $rand(0,1)$ denotes the random values ranging from 0 to 1.

[6]This chronological construction is consistent with the very nature of GC, i.e., the 'precedence' principle as we illustrated in preliminaries.

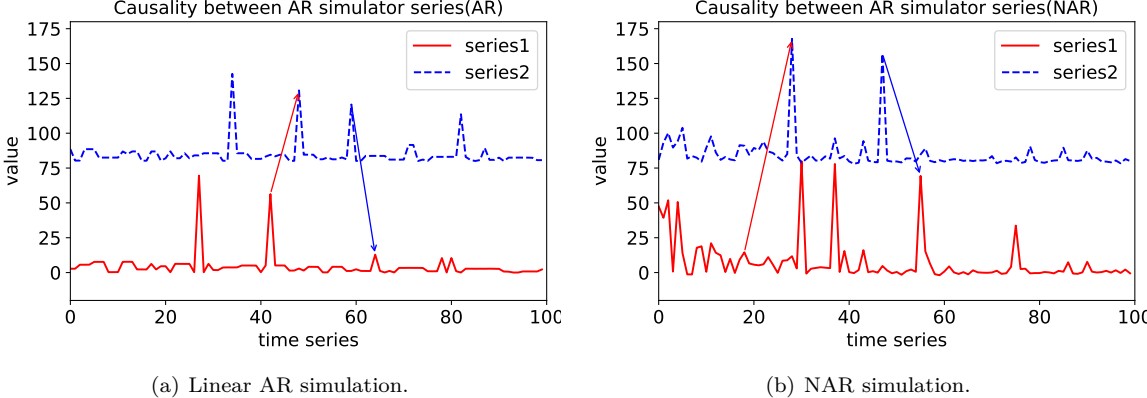

(a) Linear AR simulation.

(b) NAR simulation.

Figure 3: The partial schematic in AR/NAR experiment. The blue/red curve represents two channels of time series (series 1: $T_1$, series 2: $T_2$), and arrows represent causalities. The starting position of each causality arrow is at the point when $m_1$ ($m_2$) is abnormally large. Noting that some mutations are caused by noise instead of causalities in our construction, which are served as confounding factors.

Table 1: Recall (left) and accuracy (right) of three window-level GC methods on two AR/NAR simulation datasets.

| dataset | window size method | 10 | 20 | 30 | 100 |
|---|---|---|---|---|---|
| AR | Change-point (Aminikhanghahi & Cook, 2017) | $0.42_{(0.016)}\ 0.50_{(0.012)}$ | $0.50_{(0.017)}\ 0.50_{(0.013)}$ | $0.69_{(0.013)}\ 0.88_{(0.011)}$ | $\mathbf{0.88_{(0.005)}}\ 0.97_{(0.003)}$ |
| | Extreme causal (Bodik et al., 2021) | $0.30_{(0.018)}\ 0.47_{(0.013)}$ | $0.56_{(0.013)}\ 0.52_{(0.016)}$ | $0.76_{(0.009)}\ 0.94_{(0.010)}$ | $0.87_{(0.005)}\ 0.99_{(0.003)}$ |
| | Anomal causal (Yang et al., 2022) | $0.22_{(0.014)}\ \mathbf{0.54_{(0.013)}}$ | $0.43_{(0.018)}\ 0.52_{(0.015)}$ | $0.53_{(0.009)}\ \mathbf{0.95_{(0.010)}}$ | $0.65_{(0.004)}\ 0.96_{(0.004)}$ |
| | Rolling GC (Ming-Hsien & Chih-She, 2015) | $0.20_{(0.011)}\ 0.43_{(0.013)}$ | $0.29_{(0.010)}\ 0.52_{(0.011)}$ | $0.53_{(0.009)}\ \mathbf{0.95_{(0.010)}}$ | $0.75_{(0.002)}\ 0.99_{(0.002)}$ |
| | DWGC (ours) | $0.59_{(0.014)}\ 0.42_{(0.019)}$ | $0.60_{(0.009)}\ 0.76_{(0.010)}$ | $0.72_{(0.011)}\ 0.93_{(0.013)}$ | $0.87_{(0.007)}\ 0.99_{(0.008)}$ |
| | DWGC-CI (ours) | $\mathbf{0.84_{(0.009)}}\ 0.44_{(0.009)}$ | $\mathbf{0.89_{(0.007)}}\ \mathbf{0.79_{(0.009)}}$ | $\mathbf{0.84_{(0.005)}}\ 0.94_{(0.005)}$ | $\mathbf{0.88_{(0.005)}}\ \mathbf{0.99_{(0.001)}}$ |
| NAR | Change-point (Aminikhanghahi & Cook, 2017) | $0.39_{(0.014)}\ 0.42_{(0.013)}$ | $0.40_{(0.013)}\ 0.41_{(0.017)}$ | $0.59_{(0.014)}\ 0.80_{(0.010)}$ | $0.75_{(0.003)}\ 0.93_{(0.003)}$ |
| | Extreme causal (Bodik et al., 2021) | $0.10_{(0.019)}\ 0.37_{(0.020)}$ | $0.42_{(0.017)}\ 0.32_{(0.014)}$ | $0.56_{(0.011)}\ 0.74_{(0.011)}$ | $0.67_{(0.015)}\ 0.87_{(0.007)}$ |
| | Anomal causal (Yang et al., 2022) | $0.02_{(0.015)}\ 0.42_{(0.016)}$ | $0.35_{(0.018)}\ \mathbf{0.42_{(0.019)}}$ | $0.49_{(0.009)}\ 0.78_{(0.014)}$ | $0.58_{(0.014)}\ 0.78_{(0.016)}$ |
| | Rolling GC (Ming-Hsien & Chih-She, 2015) | $0.03_{(0.011)}\ 0.35_{(0.014)}$ | $0.25_{(0.014)}\ 0.48_{(0.017)}$ | $0.50_{(0.007)}\ \mathbf{0.85_{(0.011)}}$ | $0.64_{(0.005)}\ 0.89_{(0.003)}$ |
| | DWGC (ours) | $0.39_{(0.016)}\ 0.22_{(0.023)}$ | $0.46_{(0.010)}\ \mathbf{0.62_{(0.012)}}$ | $0.58_{(0.014)}\ 0.73_{(0.014)}$ | $\mathbf{0.76_{(0.010)}}\ 0.92_{(0.010)}$ |
| | DWGC-CI (ours) | $\mathbf{0.64_{(0.010)}}\ 0.40_{(0.011)}$ | $\mathbf{0.69_{(0.009)}}\ 0.59_{(0.010)}$ | $\mathbf{0.64_{(0.008)}}\ 0.84_{(0.009)}$ | $0.74_{(0.005)}\ \mathbf{0.94_{(0.003)}}$ |

### 5.1.2 Experimental process and result

For the two AR/NAR simulation datasets, we consider the case of $m_1, m_2 = 10$ as the beginning of the causality. To determine $\epsilon$ and the parameter of the function $h(\cdot)$, we use the 5-fold cross-validation, during which we randomly choose one hundred time points as the test set and the other four hundred as the training set. We choose the function $h(\cdot) = (\frac{6}{5} - \tanh(\cdot))$ and the causality threshold $\epsilon = 1$ (the sensitivity test of $\epsilon$ is in Appendix. D). Moreover, the rolling window intervals are taken as $\{[i * k, (i + 1)k] : i * k \in [0, L], i \in N\}$, where $k = 10, 20, 30, 100$. Moreover, in order to hold the generality, we adopt the most traditional Adam operator with the adaptive learning rate initialized as $\eta = 0.05$.

Fig. 3 shows the partial schematic of AR/NAR simulations, where the arrows mean the lag relation between the two series and we consider them as the ground truth of the dynamic causalities. Following the definition in Section 4, if $F_{2 \Longrightarrow 1}^{t,k} > 1$ or $F_{1 \Longrightarrow 2}^{t,k} > 1$ is detected on a window with at least a set of causality pairs, our causal extraction on this window is successful.

To our best knowledge, we adopt the current baseline (Bodik et al., 2021; Yang et al., 2022) of causality extraction on time series. In this case, we calculate the recall/accuracy of our methods in Table 5.1.1 after making stationary processing. As seen from the results,

1) From a vertical comparison perspective, we found that our method DWGC, especially DWGC-CI, significantly outperforms the baseline methods. Moreover, this superiority becomes more significant when the window size is smaller, as our CI matrix is specifically designed to address the instability of local windows.

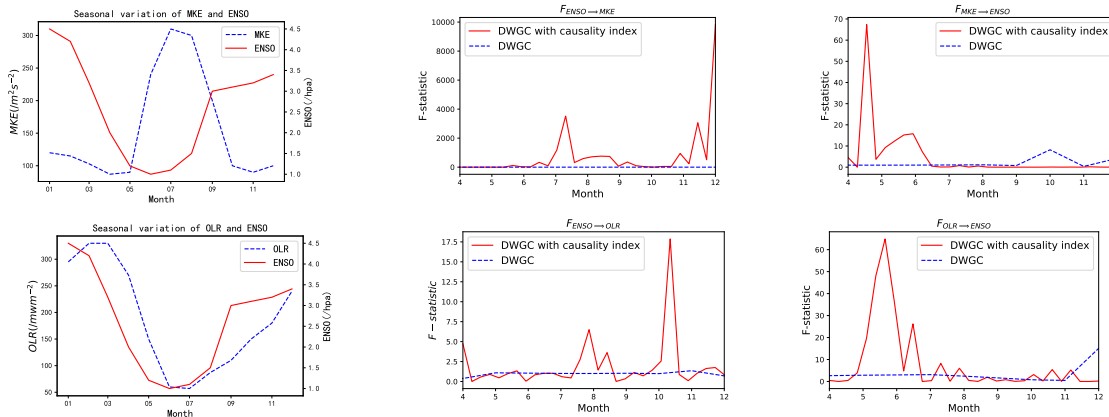

Figure 4: The experimental results on Climate Dataset for ElNiño by DWGC and DWGC-CI methods. The first column is the original data of MKE & OLR and ENSO (1992). The second and third column is the F-test result between ENSO and MKE & OLR using DWGC and DWGC-CI respectively. In all mentioned cases, our DWGC-CI method is closer to the prior knowledge of Kumar's Kumar et al. (1999) and Yasunari's Yasunari (1990).

2) With the increase in window length, both the baseline method and our DWGC method exhibit an overall improvement in accuracy and recall. This is because as the window length increases, the situation degrades into a traditional GC, and the model's performance becomes more stable, which is consistent with Theorem. 1 and Theorem. 2.
3) Furthermore, the out-performance is more significant in NAR than in AR, the potential reason is that in the NAR scenario, it is more challenging to decouple local noise and causal effects, and the performance becomes simultaneously worse. Therefore, the baseline method, which does not focus on distinguishing noise, exhibits a larger performance gap in the NAR scenario.
4) It is worth noting that this result is not consistent in all cases, as the model performance of the baseline method is unstable and biased on local windows. When we supplement variation analysis, we find that the results are basically consistent.

## 5.2 Experiment on climate dataset for ElNiño

In this part, we verify our methods on a real-world climate dataset, which has widely recognized seasonal causalities. Due to space limitation, we refer readers to Appendix. A for domain-specific knowledge.

### 5.2.1 Experimental processing and result

ENSO and Asian monsoon data in 1992 Yang et al. (2018) can be seen from the first column in Fig. 4. The first four months are taken as the training data and the rest as the testing data. In order to facilitate comparison with prior knowledge, we selected the window length $k$ as one month.

We use our model to judge the window-level causalities between ENSO and MKE & OLR every month. In the second column of Fig. 4, we show the F-statistic of every month of DWGC-CI and DWGC for two series (ENSO to MKE) and (ENSO to OLR). The x-axis is the month-time and the y-axis is the F-statistic. By DWGC, the causality between ENSO and MKE & OLR can hardly be successfully detected, not to mention the difference of causality between May-Aug (spring & summer) and Aug-Dec (autumn & winter). However, by DWGC-CI, we not only obtain the basic causalities of ENSO $\implies$ MKE and ENSO $\implies$ OLR, but also find that the causality in Sep-Dec (autumn & winter) is more significant than that in May-Aug (spring & summer). This significant causality variation detected by DWGC-CI is more consistent with the academic knowledge Kumar et al. (1999) than that by DWGC. In the third column of Fig. 4, we show the F-statistic of each month of DWGC and DWGC-CI for two series (MKE to ENSO) and (OLR to ENSO). By DWGC, we detect the causality MKE & OLR $\implies$ ENSO in Sep-Dec (autumn & winter). However, by DWGC-CI,

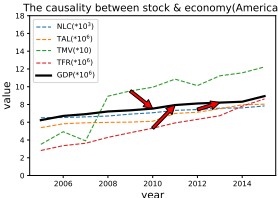 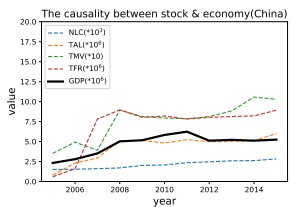 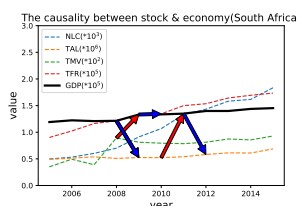

Figure 5: The causality between the stock market and economy by DWGC-CI method. Three subfigures show the trend of five variables (NLC, TAL (million), TMV (‰), TFR (million), GDP (million)) during 2007-2015 in three countries. The arrows represent causalities between different curves (the red is to GDP while the blue is from GDP). Results of our DWGC method are consistent with academic knowledge in three areas. Detailed F-statistic results are shown in Table 2,3,4.

we detect the causality MKE & OLR $\Longrightarrow$ ENSO in May-Aug (spring & summer). Compared to the DWGC method, our causalities detected by DWGC-CI are more aligned with the academic knowledge Yasunari (1990).

## 5.3 Experiment on Stock Market and Economy

In this part, we use DWGC and DWGC-CI methods to detect causality between the stock market and the economy of three representative economies: the United States, China and South Africa. The a priori knowledge is shown in Appendix. A.

### 5.3.1 Experimental processing and result

In this experiment, we choose four widely-acknowledged indexes Edou (2017) to jointly evaluate the prosperity of the stock market: the number of listed companies (NLC), the total assets listed (TAL), the total market value (TMV), and the total funds raised (TFR). On the other hand, Economic growth is measured in terms of gross domestic product (GDP). Considering the authorization limit of data sources, we selected data from three countries from 2005-2015 (5 dimensions in total) and choose 2005-2007 among them as the training data. For the sake of comparing with the prior knowledge, we set the window length $k$ as one year to observe the annual change trend of stock-economy causalities in three countries, and causal threshold $\epsilon$ = 500.

In this experiment, we test causalities between GDP and NLC, TAL, TMV, and TFR by DWGC-CI and DWGC methods, and the DWGC-CI results are illustrated by red/blue arrows in figure. 5. The results show that 1) in America, causalities NLC (2012), TMV (2009), TFR (2010) $\Longrightarrow$ GDP exist, 2) there are no specific causalities between the stock market and economy in China, and 3) bidirectional causalities TMV (2008), TAL (2010) $\Longrightarrow$ GDP, GDP $\Longrightarrow$ TAL (2008), TFR (2009,2012) exist in South Africa. However, the DWGC results don't show such a good correlation with the prior conclusions. In all, these causalities between stock & market of three areas in fig. 5 are consistent with our academic knowledge Pearce (1983); Aziz & Duenwald (2006); Ghirmay (2004) respectively by DWGC-CI, better than by DWGC. The experimental results on the stock market and economy are in table 1,2,3. In these tables, the data beyond the causal threshold $\epsilon$ = 500 are bolded, which matches the causal arrow in Fig (5).

Table 2: $F_{X \Longrightarrow GDP}$ (left) and $F_{GDP \Longrightarrow X}$ (right) of America.

| X | F-test / method | 2008 | 2009 | 2010 | 2011 | 2012 | 2013 | 2014 |
|---|---|---|---|---|---|---|---|---|
| NLC | DWGC-CI | 32.67/1.34 | 3.65/2.56 | 289.09/2.68 | 2.30/21.34 | **1269.82**/390.30 | 51.11/330.92 | 29.59/99.81 |
| | DWGC | 4.35/1.17 | 3.64/1.54 | 9.74/0.28 | 0.22/5.62 | 22.50/26.10 | 5.92/13.71 | 5.32/9.66 |
| TAL | DWGC-CI | 0.64/23.02 | 1.03/169.64 | 1.00/0.23 | 1.00/1.68 | 1.00/1.02 | 1.00/0.99 | 1.00/1.00 |
| | DWGC | 1.29/5.03 | 10.23/10.14 | 1.60/0.48 | 0.020/1.29 | 12.55/1.01 | 2.21/0.99 | 2.14/1.00 |
| TMV | DWGC-CI | 15.42/0.82 | **9480.66**/0.99 | 0.78/0.02 | 0.0036/0.024 | 91.19/0.0056 | 0.93/0.038 | 0.96/0.062 |
| | DWGC | 4.20/6.8 | 41.38/169.6 | 0.86/0.56 | 0.059/3.13 | 3.76/1.08 | 1.08/3.09 | 1.03/2.07 |
| TFR | DWGC-CI | 28.37/2.05 | 3.49/1.17 | **616.96**/2.47 | 0.078/0.13 | 20.81/0.17 | 411.65/1.67 | 11.23/5.56 |
| | DWGC | 3.94/0.07 | 3.71/0.31 | 17.10/0.43 | 0.036/1.93 | 3.68/1.47 | 9.11/10.86 | 3.23/7.12 |

Table 3: $F_{X \Longrightarrow GDP}$ (left) and $F_{GDP \Longrightarrow X}$ (right) of China.

| X | F-test / method | 2008 | 2009 | 2010 | 2011 | 2012 | 2013 | 2014 |
|---|---|---|---|---|---|---|---|---|
| NLC | DWGC-CI | 0.77/0.95 | 3.66/2.06 | 1.01/0.72 | 1.01/1.00 | 0.99/0.99 | 0.98/1.00 | 0.99/1.00 |
| | DWGC | 0.87/1.20 | 2.20/3.34 | 1.01/2.65 | 1.00/5.93 | 0.99/16.86 | 0.99/17.71 | 0.99/5.47 |
| TAL | DWGC-CI | 2.28/0.88 | 0.22/1.23 | 1.12/0.99 | 1.05/1.01 | 0.90/0.97 | 0.87/0.94 | 0.90/1.05 |
| | DWGC | 1.70/0.94 | 0.50/1.11 | 1.06/0.99 | 1.03/1.01 | 0.95/0.98 | 0.930.97 | 0.95/1.02 |
| TMV | DWGC-CI | 0.27/0.92 | 0.06/1.00 | 0.86/0.98 | 0.93/0.98 | 1.18/0.98 | 1.26/1.15 | 1.18/1.01 |
| | DWGC | 0.33/1.00 | 0.40/9.35 | 0.86/1.01 | 0.93/1.01 | 1.20/1.01 | 1.30/0.96 | 1.19/1.00 |
| TFR | DWGC-CI | 9.94/1.40 | 0.02/3.13 | 0.83/1.55 | 0.90/2.26 | 1.33/0.02 | 1.33/0.02 | 1.23/4.36 |
| | DWGC | 2.83/1.16 | 0.10/1.73 | 0.93/1.26 | 0.95/1.60 | 1.15/0.13 | 1.12/0.07 | 1.09/2.47 |

Table 4: $F_{X \Longrightarrow GDP}$ (left) and $F_{GDP \Longrightarrow X}$ (right) of South Africa.

| X | F-test / method | 2008 | 2009 | 2010 | 2011 | 2012 | 2013 | 2014 |
|---|---|---|---|---|---|---|---|---|
| NLC | DWGC-CI | 1.47/414.05 | 0.042/4.43 | 0.87/2.58 | 0.14/1.00 | 8.97/1.00 | 54.10/1.00 | 3.10/1.00 |
| | DWGC | 1.22/0.023 | 0.20/0.027 | 0.94/2.40 | 0.36/44.09 | 3.01/4.99 | 7.59/4.58 | 1.77/3.7 |
| TAL | DWGC-CI | 1.98/**8245.55** | 1.10/150.84 | **772.42**/0.45 | 0.0047/0.21 | 137.48/1.22 | 162.61/1.01 | 4.03/1.00 |
| | DWGC | 1.36/0.85 | 3.93/0.03 | 15.82/0.57 | 0.19/0.05 | 12.38/11.09 | 12.25/2.27 | 1.85/2.41 |
| TMV | DWGC-CI | **2730.82**/1.65 | 0.0018/297.02 | 0.49/0.27 | 0.0012/0.38 | 0.22/0.0039 | 0.27/2.53 | 0.36/4.72 |
| | DWGC | 53.10/1.31 | 0.41/14.96 | 0.87/0.55 | 0.25/0.63 | 0.49/0.19 | 0.51/1.47 | 0.64/1.88 |
| TFR | DWGC-CI | 1.81/1.33 | 0.45/**1205.46** | 267.35/93.50 | 0.036/0.19 | 85.53/**579.83** | 204.13/6.06 | 3.52/3.16 |
| | DWGC | 102.98/0.57 | 0.38/2.12 | 0.92/1.29 | 0.25/1.85 | 0.49/78.81 | 0.51/10.05 | 0.64/4.63 |

# 6  Conclusions and Discussions

In this paper, we focus on a new task to detect the dynamic window-level Granger causality (DWGC) between the time series. We proposed the DWGC method by conducting the F-test on comparing the window-level forecasting predictions, whereas there is demonstrated to be a granularity-efficacy tradeoff and DWGC's power of window-level GC extraction is limited. We then introduce a "causality index (CI)" method to weigh the original time series, whose purpose is to decrease the local fluctuations of the window-level F-test. The full method is DWGC-CI and is theoretically proven to have better recall rates. In the experiments on two synthetic and two real datasets, the DWGC-CI method successfully detects the window causalities, which cannot be accomplished by GC. It also outperforms DWGC, both on recall rate and accuracy rate.

This is fundamental work with many areas for future improvement and expansion. Subsequently, we will be interested in further detecting the principle of DWGC-CI on 1) window adjustment, such as dynamic rolling window intervals with different starting points, 2) weight strategy enhancement, such as assigning dynamic causality index with domain-specific knowledge, 3) graph extension, such as constructing time series causal graph combined with additional graph theory, and 4) further thoughts on the mechanism design, such as how to strike the granularity-efficacy trade-off in a more general intervention/reweighting framework.

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

## Ethics Statement

This study is conducted in accordance with the general academical principles and adhered to the corresponding ethical guidelines and standards. The research design, procedures, and data collection methods were reviewed and approved by the corresponding institutes of authors.

All participants involved in the study were informed of the study's objectives, procedures, potential risks, and benefits. Written informed consent was obtained from each participant prior to their involvement in the research. Participants were assured of the confidentiality of their personal information and were informed that they had the right to withdraw from the study at any time without any consequences.

All data collected in this study were anonymized to protect the privacy of the participants. Any identifiable information was removed, and the data were securely stored in compliance with the data protection policies. Access to the data was granted only to the research team members involved in the study.

The authors declare that they have no competing interests or conflicts of interest related to this research. The study was conducted with the utmost respect for the participants and their rights, ensuring their welfare and privacy throughout the research process.

# Supplemental Materials

## A Domain-specific knowledge in experiments

### A.1 Academic knowledge of ElNiño and monsoon

The climate academia already have the following definition on ElNiño: a climate phenomenon in the Pacific equatorial belt where the ocean and atmosphere interact with each other and lose their balance Ramage (1971), while monsoon generally eqrefers to the seasonal conversion of atmospheric circulation and precipitation in tropical and subtropical areas, which is measured by two parameters: OLR (Outgoing Longwave Radiation) and MKE (Monsoon Kinetic Energy).

The causalities between ENSO and the East Asian monsoon (MKE, OLR) have been extensively explored. Up till now, it is widely believed that the causal direction and intensity between the two has seasonal variation summarized as below:

1. The causality ENSO $\implies$ MKE & OLR exists, more intense in autumn & winter than in spring & summer.

2. The reverse causality MKE & OLR $\implies$ ENSO exists in spring & summer.

However, with the increasing instability and complexity of ENSO, the above are only observational conclusions based on meteorology knowledge and lacks highly interpretable causality analysis. In all, a prior-knowledge-fitted result in this experiment will strongly demonstrate the effectiveness of our method.

### A.2 Academic knowledge of stock market and economy in America, China and South Africa

The relationship between the development of the stock market and economic growth has long been a hot topic in financial circles. It's one of the most classical cases where a mutual causal-and-effect relationship exists. These causalities are hard to be captured since they show different characteristics in various social environments and economic development stages.

In the world, the United States, China and South Africa, as three highly representative countries in the Americas, Asia and Africa, contain the empirical knowledge of stock-economy causality summarized as follows:

1 the United States: Pearce Pearce (1983) showed that the stock market was a long-term stability indicator of economic growth, i.e., Stock Market $\Rightarrow$ Economy.

2 China: Aziz Aziz & Duenwald (2006) used the panel data method, and found that China's stock market development had little impact on GDP growth during the post-1978 eqreform period.

3 South Africa: Ghirmay Ghirmay (2004) studied the causal relationship between financial development and economic growth in 13 sub-Saharan African countries by bivariate VAR model, results showed that financial development promoted economic growth in eight countries, and among which there were bidirectional causalities in six countries, i.e., Economy $\Rightarrow$ Stock Market.

## B Proof of Theorem. 1

*Proof.* Without loss of generation, we can choose $\epsilon = 1$, and the approximate Gaussian distribution on each time point shares a similar variance as in Assumption. 1. $Y_{i,t}$ is partitioned into **o**riginal real data $Y_{i,t}^o$ and **n**oise $Y_{i,t}^n$ as

$$Y_{i,t} = Y_{i,t}^o + Y_{i,t}^n. \tag{8}$$

We assume that $\widehat{Y_{i,t}} \sim \mathcal{N}(Y_{i,t}^o, \sigma^2)$, $\widehat{Y_{i|j,t}} \sim \mathcal{N}(Y_{i,t}^o, \epsilon_0^2 \sigma^2)$, where $\epsilon_0 \in (0,1)$, $\sigma$ are constants. Moreover, the Gaussian noise satisfies $Y_{i,t}^n \sim \mathcal{N}(0, \sigma_0)$. In Equation. 3 , if we denote $\frac{\sum_q (\hat{Y}_{i,q} - Y_{i,q})^2}{\sum_q (\hat{Y}_{i|j,q} - Y_{i,q})^2} = \frac{\sigma^2 + \sigma_0^2}{\epsilon_0^2 \sigma^2 + \sigma_0^2} X$, then we conclude that $X$ satisfies the $\mathcal{F}$ distribution. Its density probability function is

$$f(x) = \begin{cases} 0, x \le 0 \\ \frac{1}{\mathcal{B}(\frac{k}{2}, \frac{k}{2})} x^{\frac{k}{2}-1}(1+x)^{-k}, x > 0. \end{cases} \tag{9}$$

In our proof, we choose the threshold 1. On this basis, we have

$$\begin{aligned} P(F_{j \Longrightarrow i}^{t,k} > 1) &= \int_{\frac{\epsilon_0^2 \sigma^2 + \sigma_0^2}{\sigma^2 + \sigma_0^2}}^{+\infty} \frac{1}{\mathcal{B}(\frac{k}{2}, \frac{k}{2})} x^{\frac{k}{2}-1}(1+x)^{-k} dx \\ &= \frac{1}{\mathcal{B}(\frac{k}{2} \frac{k}{2})} \int_{x_0}^1 x^{\frac{k}{2}-1}(1-x)^{\frac{k}{2}-1} dx, \quad (x_0 := \frac{\epsilon_0^2 \sigma^2 + \sigma_0^2}{(\epsilon_0^2+1)\sigma^2 + 2\sigma_0^2} \in (0, \frac{1}{2})) \\ &=: 1 - I_{x_0}(\frac{k}{2}, \frac{k}{2}) \end{aligned} \tag{10}$$

We introduce the following lemma:

**Lemma 1.** *If* $x_0 \in (0, \frac{1}{2})$, *then*

$$\begin{aligned} I_{x_0}(\frac{k}{2}, \frac{k}{2}) &= \frac{\Gamma(k)}{\Gamma(\frac{k}{2}+1)\Gamma(\frac{k}{2})}(1-2x_0)x_0^{\frac{k}{2}}(1-x_0)^{\frac{k}{2}} + I_{x_0}(\frac{k+2}{2}, \frac{k+2}{2}) \\ &\ge I_{x_0}(\frac{k+2}{2}, \frac{k+2}{2}). \end{aligned} \tag{11}$$

We make contradictions. If there is $k = i_0$ such that $I_{x_0}(\frac{i_0}{2}, \frac{i_0}{2}) > I_{x_0}(\frac{i_0-1}{2}, \frac{i_0-1}{2})$. Then we have

$$\begin{aligned} &I_{x_0}(\frac{i_0+2}{2}, \frac{i_0+2}{2}) - I_{x_0}(\frac{i_0+1}{2}, \frac{i_0+1}{2}) \\ \ge &I_{x_0}(\frac{i_0}{2}, \frac{i_0}{2}) - I_{x_0}(\frac{i_0-1}{2}, \frac{i_0-1}{2}) + \frac{\Gamma(k-1)(1 + \frac{2(k-1)}{k+1}x_0)x_0^{\frac{k-1}{2}}(1-x_0)^{\frac{k-1}{2}}}{\Gamma(\frac{k-1}{2}+1)\Gamma(\frac{k-1}{2})} \\ &- \frac{\Gamma(k)(1 + \frac{2k}{k+2}x_0)x_0^{\frac{k}{2}}(1-x_0)^{\frac{k}{2}}}{\Gamma(\frac{k}{2}+1)\Gamma(\frac{k}{2})} \\ = &I_{x_0}(\frac{i_0}{2}, \frac{i_0}{2}) - I_{x_0}(\frac{i_0-1}{2}, \frac{i_0-1}{2}). \end{aligned} \tag{12}$$

Hence we have $0 = \lim_{k \to +\infty} |I_{x_0}(\frac{k+1}{2}, \frac{k+1}{2}) - I_{x_0}(\frac{k}{2}, \frac{k}{2})| \ge I_{x_0}(\frac{i_0}{2}, \frac{i_0}{2}) - I_{x_0}(\frac{i_0-1}{2}, \frac{i_0-1}{2}) > 0$. It's contradictive! Therefore, we have $\forall k, I_{x_0}(\frac{k}{2}, \frac{k}{2}) > I_{x_0}(\frac{k+1}{2}, \frac{k+1}{2})$. Thus

$$\forall k, \ P(F_{j \Longrightarrow i}^{t,k} > 1) < P(F_{j \Longrightarrow i}^{t,k+1} > 1). \tag{13}$$

Moreover, according to lemma. 1, we further have

$$\begin{aligned} 0 < \mathbb{P}(F_{j \Longrightarrow i}^{t,k_2} > 1) - \mathbb{P}(F_{j \Longrightarrow i}^{t,k_1} > 1) &= \sum_{i=k_1}^{k_2} [I_{x_0}(\frac{i}{2}, \frac{i}{2}) - I_{x_0}(\frac{i+1}{2}, \frac{i+1}{2})] \\ &= \sum_{i=k_1}^{k_2} \mathcal{O}\left( \frac{\Gamma(i)}{\Gamma(\frac{i}{2}+1)\Gamma(\frac{i}{2})}(1-2x_0)x_0^{\frac{i}{2}}(1-x_0)^{\frac{i}{2}} \right) \\ &< C\left( (\frac{1}{4})^{\frac{k_1}{2}}(k_2 - k_1) \right). \end{aligned} \tag{14}$$

Proved.

$\square$

### B.1 The proof of lemma. 1

*Proof.* Via integration by parts, we have

$$
\begin{aligned}
& I_{x_0}(p, q) - I_{x_0}(p+1, q) \\
&= \frac{\int_0^{x_0} x^{p-1}(1-x)^{q-1}}{\mathcal{B}(p, q)} - \frac{\int_0^{x_0} x^p(1-x)^{q-1}}{\mathcal{B}(p+1, q)} \\
&= \frac{1}{\mathcal{B}(p, q)}[\int_0^{x_0} x^{p-1}(1-x)^{q-1}dx - \frac{q+p}{p}\int_0^{x_0} x^p(1-x)^{q-1}dx] \\
&= \frac{1}{\mathcal{B}(p, q)}[\int_0^{x_0} x^{p-1}(1-x)^q dx - \frac{q}{p}\int_0^{x_0} x^p(1-x)^{q-1}dx] \\
&= \frac{1}{\mathcal{B}(p, q)}[\int_0^{x_0} x^{p-1}(1-x)^q dx - (\frac{1}{p}x^p(1-x)^q \mid_0^{x_0} + \int_0^{x_0} x^{p-1}(1-x)^q dx)] \\
&= \frac{\Gamma(p+q)}{\Gamma(p+1)\Gamma(q)}x_0^p(1-x_0)^q.
\end{aligned}
\tag{15}
$$

If we choose $p = \frac{k}{2}, q = \frac{k}{2}$, then

$$
I_{x_0}(\frac{k}{2}, \frac{k}{2}) = \frac{\Gamma(k)}{\Gamma(\frac{k}{2}+1)\Gamma(\frac{k}{2})}x_0^{\frac{k}{2}}(1-x_0)^{\frac{k}{2}} + I_{x_0}(\frac{k+2}{2}, \frac{k}{2})
\tag{16}
$$

On the other hand, we also have

$$
\begin{aligned}
I_{x_0}(p, q) &= 1 - \frac{\int_{x_0}^1 x^{p-1}(1-x)^{q-1}dx}{\mathcal{B}(p, q)} \\
&= 1 - \frac{\int_0^{1-x_0} x^{q-1}(1-x)^{p-1}dx}{\mathcal{B}(p, q)} \\
&= 1 - I_{1-x_0}(q, p) \\
&\overset{*}{=} 1 - I_{1-x_0}(q+1, p) - \frac{\Gamma(p+q)}{\Gamma(q+1)\Gamma(p)}(1-x_0)^q x_0^p \\
&= I_{x_0}(p, q+1) - \frac{\Gamma(p+q)}{\Gamma(q+1)\Gamma(p)}(1-x_0)^q x_0^p,
\end{aligned}
\tag{17}
$$

where $*$ denotes 15. If we choose $p = \frac{k+2}{2}, q = \frac{k}{2}$, then

$$
I_{x_0}(\frac{k+2}{2}, \frac{k}{2}) = -\frac{\Gamma(k+1)}{\Gamma(\frac{k}{2}+1)\Gamma(\frac{k+2}{2})}x_0^{\frac{k+2}{2}}(1-x_0)^{\frac{k}{2}} + I_{x_0}(\frac{k+2}{2}, \frac{k+2}{2}).
\tag{18}
$$

Combined 16 and 18, and due to $x_0 \in (0, \frac{1}{2})$, we conclude that

$$
\begin{aligned}
I_{x_0}(\frac{k}{2}, \frac{k}{2}) &= \frac{\Gamma(k)}{\Gamma(\frac{k}{2}+1)\Gamma(\frac{k}{2})}(1-2x_0)x_0^{\frac{k}{2}}(1-x_0)^{\frac{k}{2}} + I_{x_0}(\frac{k+2}{2}, \frac{k+2}{2}) \\
&> I_{x_0}(\frac{k+2}{2}, \frac{k+2}{2}).
\end{aligned}
\tag{19}
$$

Proved. $\square$

## C   The proof of Corollary. 1

*Proof.* We have

$$
\begin{aligned}
\mid P(F^{t,k}_{j \Longrightarrow i} > 1) - P(F^{t,L}_{j \Longrightarrow i} > 1) \mid &= \sum_{i=k}^{L} [I_{x_0}(\frac{i}{2}, \frac{i}{2}) - I_{x_0}(\frac{i+1}{2}, \frac{i+1}{2})] \\
&= \sum_{i=k}^{L} \mathcal{O}\left( \frac{\Gamma(i)}{\Gamma(\frac{i}{2}+1)\Gamma(\frac{i}{2})} (1 - 2x_0) x_0^{\frac{i}{2}} (1 - x_0)^{\frac{i}{2}} \right) \\
&= \mathcal{O}\left( (\frac{1}{4})^{\frac{k}{2}} (L - k) \right),
\end{aligned}
\tag{20}
$$

$\mathcal{O}$ is the asymptotic notation denoting the orders of approximation following Hardy (1924). In conclusion, if GC exists, i.e., $\epsilon_0 \in (0,1)$, the recall rate of DWGC is positively related to the window length $k$ and is convergent to 1 when $k \to +\infty$. Proved. □

# D Proof of Theorem 2

Analogous to the above section, the re-weighted. $Y_{i,t}^{\Phi}$ is partitioned as

$$Y_{i,t}^{\Phi} := \Phi_t Y_{i,t} = \Phi_t Y_{i,t}^{o} + \Phi_t Y_{i,t}^{n}. \tag{21}$$

On this basis, we assume $\widehat{Y_{i,t}^{\Phi}} \sim \mathcal{N}(\Phi_t Y_{i,t}^{o}, \Phi_t^2 \sigma^2)$, $\widehat{Y_{i|j,t}^{\Phi}} \sim \mathcal{N}(\Phi_t Y_{i,t}^{o}, \Phi_t^2 \epsilon_0^2 \sigma^2)$, $\Phi_t \widehat{Y_{i,t}^{n}} \sim \mathcal{N}(0, \Phi_t^2 \sigma_0^2)$. Moreover, for simplification, we assume $\Phi_{t_i} \neq \Phi_{t_j}, t_i \neq t_j$, or else a small perturbation is applied. According to the generalized chi-square distribution in Hammarwall et al. (2008), the summarization satisfies the following probability density distribution:

$$
\begin{aligned}
\sum_{q=t}^{t+k-1} (\hat{Y}_{i,q}^{\Phi} - Y_{i,q}^{\Phi})^2 &\sim f_1(x; \Phi_t, ...\Phi_{t+k-1}) \\
&:= \sum_{q=t}^{t+k-1} \frac{e^{-\frac{x}{\Phi_q^2(\sigma^2 + \sigma_0^2)}}}{\Phi_q^2(\sigma^2 + \sigma_0^2) \prod_{j=t,j\neq q}^{t+k-1}\left(1 - \frac{\Phi_j^2}{\Phi_q^2}\right)} \text{ , for } x \geq 0. \\
\sum_{q=t}^{t+k-1} (\hat{Y}_{i|j,q}^{\Phi} - Y_{i,q}^{\Phi})^2 &\sim f_2(x; \Phi_t, ...\Phi_{t+k-1}) \\
&:= \sum_{q=t}^{t+k-1} \frac{e^{-\frac{x}{\Phi_q^2(\epsilon_0^2\sigma^2 + \sigma_0^2)}}}{\Phi_q^2(\epsilon_0^2\sigma^2 + \sigma_0^2) \prod_{j=t,j\neq q}^{t+k-1}\left(1 - \frac{\Phi_j^2}{\Phi_q^2}\right)} \text{ , for } x \geq 0.
\end{aligned}
\tag{22}
$$

On this basis, we construct $\{\Phi_t, ...\Phi_{t+k-1}\}$ to make $P(\frac{\sum_q(\hat{Y}_{i,q}^{\Phi} - Y_{i,q}^{\Phi})^2}{\sum_q(\hat{Y}_{i|j,q}^{\Phi} - Y_{i,q}^{\Phi})^2} > 1) > P(\frac{\sum_q(\hat{Y}_{i,q} - Y_{i,q})^2}{\sum_q(\hat{Y}_{i|j,q} - Y_{i,q})^2} > 1)$. In this sense, we equivalently transform the Theorem. 2 to the following lemma:

**Lemma 2.** *If we have* $\sum_{q_1=t}^{t+k-1}\sum_{q_2=t}^{t+k-1} \frac{1}{(c_0\frac{\Phi_{q_2}^2}{\Phi_{q_1}^2}+1)\prod_{j=t,j\neq q_1}^{t+k-1}(1-\frac{\Phi_j^2}{\Phi_{q_1}^2})\prod_{j=t,j\neq q_2}^{t+k-1}(1-\frac{\Phi_j^2}{\Phi_{q_2}^2})} > \int_{c_0}^{+\infty} \frac{1}{\mathcal{B}(\frac{k}{2},\frac{k}{2})}x^{\frac{k}{2}-1}(1+x)^{-k}dx$, *where* $c_0 = \frac{\epsilon_0^2\sigma^2+\sigma_0^2}{\sigma^2+\sigma_0^2}$. *Then the recall rate of DWGC-CI outperforms the DWGC's in window length* $k$.

Notice that the left part in the lemma. 2 holds the property

$$
\sup_{\{\Phi_t, ...\Phi_{t+k-1}\}\in R^k} \sum_{q_1=t}^{t+k-1}\sum_{q_2=t}^{t+k-1} \frac{1}{(c_0\frac{\Phi_{q_2}^2}{\Phi_{q_1}^2}+1)\prod_{j=t,j\neq q_1}^{t+k-1}(1-\frac{\Phi_j^2}{\Phi_{q_1}^2})\prod_{j=t,j\neq q_2}^{t+k-1}(1-\frac{\Phi_j^2}{\Phi_{q_2}^2})} = +\infty, \tag{23}
$$

thus such set $\{\Phi_t, ...\Phi_{t+k-1}\}$ in lemma. 2 naturally exists. On this basis, following the lemma. 2, the regularization term can be designed as

$$
\left\| \sum_{q_1=t}^{t+k-1}\sum_{q_2=t}^{t+k-1} \frac{1}{(c_0\frac{\Phi_{q_2}^2}{\Phi_{q_1}^2}+1)\prod_{j=t,j\neq q_1}^{t+k-1}(1-\frac{\Phi_j^2}{\Phi_{q_1}^2})\prod_{j=t,j\neq q_2}^{t+k-1}(1-\frac{\Phi_j^2}{\Phi_{q_2}^2})} - \int_{c_0}^{+\infty} \frac{1}{\mathcal{B}(\frac{k}{2},\frac{k}{2})}x^{\frac{k}{2}-1}(1+x)^{-k}dx \right\|_2. \tag{24}
$$

### D.1 The proof of lemma. 2

*Proof.*

$$P(\frac{\sum_q (\hat{Y}_{i,q} - Y_{i,q})^2}{\sum_q (\hat{Y}_{i|j,q} - Y_{i,q})^2} > 1)$$

$$= \int_{c_0}^{+\infty} \frac{1}{\mathcal{B}(\frac{k}{2}, \frac{k}{2})} x^{\frac{k}{2}-1}(1+x)^{-k} dx$$

$$\overset{*}{\leq} \sum_{q_1=t}^{t+k-1} \sum_{q_2=t}^{t+k-1} \frac{1}{(c_0 \frac{\Phi_{q_2}^2}{\Phi_{q_1}^2} + 1)\prod_{j=t,j\neq q_1}^{t+k-1}(1 - \frac{\Phi_j^2}{\Phi_{q_1}^2})\prod_{j=t,j\neq q_2}^{t+k-1}(1 - \frac{\Phi_j^2}{\Phi_{q_2}^2})}$$

$$= \sum_{q_1=t}^{t+k-1} \sum_{q_2=t}^{t+k-1} \frac{\frac{1}{\frac{1}{\Phi_{q_1}^2(\sigma^2+\sigma_0^2)} + \frac{1}{\Phi_{q_2}^2(\epsilon_0^2\sigma^2+\sigma_0^2)}}}{\prod_{j=t,j\neq q_1}^{t+k-1}(1 - \frac{\Phi_j^2}{\Phi_{q_1}^2})\Phi_{q_2}^2(\epsilon_0^2\sigma^2 + \sigma_0^2)\prod_{j=t,j\neq q_2}^{t+k-1}}$$

$$= \int_0^{+\infty} \sum_{q_1=t}^{t+k-1} \sum_{q_2=t}^{t+k-1} \frac{e^{-\frac{x}{\Phi_{q_1}^2(\sigma^2+\sigma_0^2)} - -\frac{x}{\Phi_{q_2}^2(\epsilon_0^2\sigma^2+\sigma_0^2)}}}{\prod_{j=t,j\neq q_1}^{t+k-1}(1 - \frac{\Phi_j^2}{\Phi_{q_1}^2})\Phi_{q_2}^2(\epsilon_0^2\sigma^2 + \sigma_0^2)\prod_{j=t,j\neq q_2}^{t+k-1}} dx \qquad (25)$$

$$= \int_0^{+\infty} \int_x^{+\infty} \sum_{q=t}^{t+k-1} \frac{e^{-\frac{m}{\Phi_q^2(\sigma^2+\sigma_0^2)}}}{\Phi_q^2(\sigma^2 + \sigma_0^2)\prod_{j=t,j\neq q}^{t+k-1}\left(1 - \frac{\Phi_j^2}{\Phi_q^2}\right)} dm$$

$$\sum_{q=t}^{t+k-1} \frac{e^{-\frac{x}{\Phi_q^2(\epsilon_0^2\sigma^2+\sigma_0^2)}}}{\Phi_q^2(\epsilon_0^2\sigma^2 + \sigma_0^2)\prod_{j=t,j\neq q}^{t+k-1}\left(1 - \frac{\Phi_j^2}{\Phi_q^2}\right)} dx$$

$$= \int_0^{+\infty} \int_x^{+\infty} f_2(m)f_1(x) dm dx$$

$$= P(\frac{\sum_q (\hat{Y}_{i,q}^\Phi - Y_{i,q}^\Phi)^2}{\sum_q (\hat{Y}_{i|j,q}^\Phi - Y_{i,q}^\Phi)^2} > 1)$$

Here $*$ is from the inequality in lemma. 2. Hence we proved $P(\frac{\sum_q (\hat{Y}_{i,q}^\Phi - Y_{i,q}^\Phi)^2}{\sum_q (\hat{Y}_{i|j,q}^\Phi - Y_{i,q}^\Phi)^2} > 1) > P(\frac{\sum_q (\hat{Y}_{i,q} - Y_{i,q})^2}{\sum_q (\hat{Y}_{i|j,q} - Y_{i,q})^2} > 1)$. $\qquad\square$

## E  Sensitivity analysis

### E.1  Sliding window size selection

The impact of window size selection is shown in Tab. 5.

Table 5: To illustrate the impact of the window size selection, we add an experimental comparison using the American data of 2008 in the stock-and-market experiment. We change the sliding window size to a month and the results are as follows. Compared with the results in the paper, the detected causality does not completely fit with the prior knowledge and possibly misses some part of the dynamic causality directions.

| X | F-test / method | Jan-Feb | Mar-Apr | May-Jun | Jul-Aug | Sept-Oct | Nov-Dec |
|---|---|---|---|---|---|---|---|
| NLC | DWGC-CI | 21.67/12.30 | 38.6/22.51 | 28.09/2.60 | 7.20/34.54 | 12.62/39.36 | 34.76/53.65 |
| | DWGC | 32.64/43.65 | 5.32/5.32 | 6.43/54.32 | 5.23/46.23 | 53.23/54.23 | 54.32/45.32 |
| TAL | DWGC-CI | 3.21/32.12 | 32.12/54.23 | 43.23/0.54 | 32.11/14.22 | 1.23/2.34 | 32.43/0.53 |
| | DWGC | 43.12/43.23 | 34.21/32.12 | 54.23/0.43 | 0.32/54.23 | 54.23/56.65 | 54.23/0.43 |
| TMV | DWGC-CI | 43.23/54.23 | 43.23/0.65 | 0.76/0.89 | 0.43/0.67 | 91.54/54.23 | 0.76/0.32 |
| | DWGC | 54.34/23.32 | 54.32/34.23 | 43.23/43.54 | 12.23/45.23 | 34.12/35.32 | 4.12/4.23 |
| TFR | DWGC-CI | 47.32/56.09 | 4.23/45.21 | 54.23/56.64 | 54.23/65.23 | 43.23/1.24 | 43.23/23.12 |
| | DWGC | 43.23/49.34 | 2.24/43.23 | 34.23/0.34 | 1.24/1.65 | 9.34/4.34 | 4.32/65.23 |

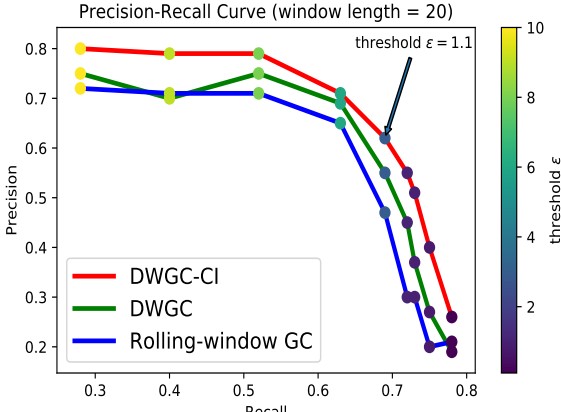

## E.2 Precision-recall rate (threshold selection)

This is an auxiliary experiment for our AR/NAR simulation. In this section, we set window length $= 20$ in the NAR case, for instance, to test the correlation between the causality threshold ($\epsilon$) and the performance (precision and recall). The experimental results show that our method DWGC-CI is robust to the selection of epsilon values and consistently outperforms DWGC and Rolling-window GC. Moreover, the difference between the performances of various methods is most significant at $\epsilon = 1.1$.

**Very few false-positive rates without GC ground truth** If the data has no GC relation, our DWGC-CI captures very few false positives. We add the following result to the synthetic data experiment: two series with no correlation (and certainly no GC) are constructed as $T_{i,t} = rand(0,1), i = 0, 2, t = 0\ 1000$ , where is a random variable ranging from to . We find that the probability of being judged as GC by DWGC-CI is relatively low at all window sizes, as follows. When there is no correlation between the two series, the false-positive probability is relatively low.

| Sliding window size | 10 | 20 | 30 | 100 |
|---|---|---|---|---|
| False-positive rate of DWGC | 0.142 | 0.190 | 0.072 | 0.032 |
| False-positive rate of DWGC-CI | **0.041** | **0.024** | **0.039** | **0.00** |

Table 6: Very few false-positive rates without GC ground truth of DWGC-CI.

## E.3 Additional analysis on the training process

In this section, we showcase three interesting issues: 1) What about conducting CI matrix updating on all points instead of the spurious windows where causality exists as in Eqn 5? 2) What about choosing difference form of $h(\cdot)$ function, such as $\alpha - tanh(\cdot)$ with different $\alpha$? and 3) What about the NAR model is not sufficiently trained?
We take the NAR case in the synthetic experiment for instance, and the result is summarized as in the following table:

1) Although the operation of updating CI for the whole graph is still effective, its performance is simultaneously poorer than the original form.
2) DWGC-CI will significantly fail when $\alpha$ is far away than 1 since our model would be potentially unstable when the total data scale is not equal to one, which is consistent with the original form in our main text.
3) Unlike the sensitivity of alpha in the second point, DWGC-CI is not so sensitive to whether the NAR model is well-trained or not. This is because we want to examine the F-statistics value, which measures

Table 7: The extension from the bi-channel case to the multi-channel case (recall (left) and accuracy (right)).

| dataset / method | window size | 10 | 20 | 30 | 100 |
|---|---|---|---|---|---|
| DWGC-CI (original, $\alpha = 1.1$) | | $0.59_{(0.013)}$ $0.56_{(0.014)}$ | $0.94_{(0.010)}$ $0.76_{(0.014)}$ | $0.87_{(0.012)}$ $0.95_{(0.004)}$ | $0.88_{(0.015)}$ $0.98_{(0.002)}$ |
| DWGC-CI ($\Phi$ on the whole graph) | | $0.55_{(0.043)}$ $0.43_{(0.054)}$ | $0.93_{(0.013)}$ $0.76_{(0.016)}$ | $0.80_{(0.011)}$ $0.78_{(0.055)}$ | $0.80_{(0.015)}$ $0.80_{(0.012)}$ |
| DWGC-CI($\alpha = 10$) | | $0.32_{(0.043)}$ $0.65_{(0.065)}$ | $0.76_{(0.054)}$ $0.76_{(0.054)}$ | $0.76_{(0.042)}$ $0.65_{(0.044)}$ | $0.58_{(0.025)}$ $0.98_{(0.002)}$ |
| DWGC-CI (10% time of NAR training) | | $0.56_{(0.017)}$ $0.56_{(0.020)}$ | $0.90_{(0.020)}$ $0.74_{(0.024)}$ | $0.80_{(0.017)}$ $0.90_{(0.007)}$ | $0.84_{(0.015)}$ $0.98_{(0.005)}$ |

the difference between two fittings using the same model rather than necessarily achieving the best fit in a specific fitting.

## F    Additional discussion

### F.1    Concept drift

Besides, we can also treat this problem given "concept drift" Tsymbal (2004), which means that "changes in the hidden context can induce more or less radical changes in the target concept" Tsymbal (2004). "Concept drift" falls into two categories: sudden (instantaneous) and gradual. The traditional GC only extracts the causal information from the latter and ignores the former, while our DWGC-CI method can extract both.

### F.2    Back door criteria

The premise of 2 is that series channels $i$ and $j$ meet the "backdoor condition" Pearl & Robins (1995). That is, we should first exclude common confounding from other channels that both effect series $i$ and $j$. Moreover, in a more strict sense, Granger causality is considered as a non-causal "precedence" Pearl (2018), according to the informal fallacy "Post hoc ergo propter hoc" Woods & Walton (1977) which means that "after this, therefore because of this". It is a flaw inherent in GC's approach itself. Thus the objective of our paper is to generalise the existing GC rather than sticking to such philosophical dilemmas.

Despite of this, we still perform an extension from the bi-level case to the multi-level case. We have expanded the 2-dimensional experiments to 20 dimensions. Our approach consists of two steps: 1) performing a 2-dimensional operation on any two dimensions to examine the window-level causal relationships, and 2) for all elements where $Y_{i_1}(t_{i_1}) < -Y_j(t_j) - > Y_{i_2}(t_{i2})$ is detected, we predict $Y_{i_1}$ and $Y_{i_2}$ sequences again after removing the information of $Y_j$ to eliminate the potential influence of confounders on causal identification. With this basic operation, our DWGC-CI method significantly outperforms all baselines. The causal graph upon multi-dimensional time series is defined as

$$T_i(t) = \text{Re}\left(\sqrt{T_{j,t-lag}^2 - 1}\right) + n(t), i, j \in 1, 2, \ldots M,$$

which is naturally derived from our main text. Here $M$ is the total sum of the number of channels. *Please note that the multidimensional approaches discussed here are only provided as examples and are not exhaustive. There are other more complex and sophisticated methods available such as [1]. The application and comparison of these methods can be considered as future work.

### F.3    More comparison with other methods

Compared with change-point detection and model parameters smoothing, window-level causality detection has the following advantages:
a) Finer time scales: Window-level causality detection can analyze and detect causal relationships on finer time scales, whereas change-point detection usually analyzes based on the entire time series and cannot provide the same fine temporal resolution.
b)Localized analysis: Window-level causality detection divides the time series into consecutive windows and

Table 8: The extension from the bi-channel case to the multi-channel case (recall (left) and accuracy (right)).

| dataset | method \ window size | 10 | 20 | 30 | 100 |
|---|---|---|---|---|---|
| AR | Change-point (Aminikhanghahi & Cook, 2017) | $0.46_{(0.018)}$ $0.55_{(0.013)}$ | $0.55_{(0.019)}$ $0.45_{(0.014)}$ | $0.76_{(0.014)}$ $\mathbf{0.98_{(0.012)}}$ | $0.97_{(0.006)}$ $0.97_{(0.003)}$ |
|  | Extreme causal (Bodik et al., 2021) | $0.40_{(0.027)}$ $0.57_{(0.019)}$ | $0.52_{(0.019)}$ $0.62_{(0.024)}$ | $0.71_{(0.013)}$ $0.93_{(0.015)}$ | $0.82_{(0.010)}$ $0.94_{(0.013)}$ |
|  | Anomal causal (Yang et al., 2022) | $0.17_{(0.021)}$ $\mathbf{0.70_{(0.018)}}$ | $0.46_{(0.027)}$ $0.71_{(0.023)}$ | $0.58_{(0.013)}$ $0.94_{(0.015)}$ | $0.61_{(0.012)}$ $0.94_{(0.012)}$ |
|  | Rolling GC (Ming-Hsien & Chih-She, 2015) | $0.15_{(0.016)}$ $0.65_{(0.019)}$ | $0.39_{(0.015)}$ $0.71_{(0.016)}$ | $0.58_{(0.013)}$ $0.93_{(0.015)}$ | $0.63_{(0.016)}$ $0.94_{(0.005)}$ |
|  | DWGC (ours) | $0.68_{(0.019)}$ $0.53_{(0.028)}$ | $0.71_{(0.013)}$ $\mathbf{0.87_{(0.015)}}$ | $0.83_{(0.016)}$ $0.94_{(0.020)}$ | $0.85_{(0.013)}$ $0.98_{(0.019)}$ |
|  | DWGC-CI (ours) | $\mathbf{0.93_{(0.013)}}$ $0.66_{(0.014)}$ | $\mathbf{0.99_{(0.010)}}$ $0.86_{(0.014)}$ | $\mathbf{0.99_{(0.012)}}$ $0.95_{(0.005)}$ | $\mathbf{0.99_{(0.010)}}$ $\mathbf{1.00_{(0.012)}}$ |
| NAR | Change-point (Aminikhanghahi & Cook, 2017) | $0.40_{(0.017)}$ $0.50_{(0.023)}$ | $0.48_{(0.023)}$ $0.40_{(0.012)}$ | $0.70_{(0.034)}$ $0.90_{(0.018)}$ | $0.87_{(0.016)}$ $0.89_{(0.005)}$ |
|  | Extreme causal (Bodik et al., 2021) | $0.15_{(0.021)}$ $0.57_{(0.022)}$ | $0.57_{(0.019)}$ $0.47_{(0.017)}$ | $0.73_{(0.015)}$ $0.87_{(0.011)}$ | $0.75_{(0.013)}$ $0.87_{(0.016)}$ |
|  | Anomal causal (Yang et al., 2022) | $0.01_{(0.017)}$ $\mathbf{0.70_{(0.018)}}$ | $0.49_{(0.020)}$ $0.68_{(0.021)}$ | $0.65_{(0.011)}$ $0.86_{(0.016)}$ | $0.70_{(0.015)}$ $0.87_{(0.016)}$ |
|  | Rolling GC (Ming-Hsien & Chih-She, 2015) | $0.07_{(0.015)}$ $0.49_{(0.018)}$ | $0.31_{(0.017)}$ $0.72_{(0.020)}$ | $0.55_{(0.011)}$ $0.92_{(0.014)}$ | $0.60_{(0.010)}$ $0.92_{(0.014)}$ |
|  | DWGC (ours) | $0.44_{(0.018)}$ $0.12_{(0.025)}$ | $0.53_{(0.013)}$ $\mathbf{0.82_{(0.016)}}$ | $0.68_{(0.015)}$ $0.81_{(0.022)}$ | $0.73_{(0.012)}$ $0.87_{(0.002)}$ |
|  | DWGC-CI (ours) | $\mathbf{0.59_{(0.013)}}$ $0.56_{(0.014)}$ | $\mathbf{0.94_{(0.010)}}$ $0.76_{(0.014)}$ | $\mathbf{0.87_{(0.012)}}$ $\mathbf{0.95_{(0.004)}}$ | $\mathbf{0.88_{(0.015)}}$ $\mathbf{0.98_{(0.002)}}$ |

performs causality analysis within each window. This localized analysis can more accurately capture local variations and causality in the time series without being affected by the entire time series.

c) Better sensitivity and specificity: Window-level causality detection can analyze the data within each window, targeting causality judgments to the characteristics of the data within a specific window. This targeted analysis improves the sensitivity and specificity of the detection algorithm and reduces false positives and misses.

d) Most importantly, adaptation to dynamic changes: Window-level causality detection can flexibly adapt to dynamic changes in the time series, and **can identify complex relationships between different change points**. The window size can be adjusted as needed to accommodate potential causal changes in different time periods. This dynamic adaptability makes window-level causality detection more applicable and robust. In sum, window-level causality detection has more advantages than variable-point detection in terms of time scale, analysis accuracy, sensitivity, and adaptability. The experiment also demonstrates the superiority of DWGC-CI over traditional change-point detection.

## F.4 Justification of assumption

**Justification of Gaussian assumption in the theoretical part.** In the context of time series modeling, the Regressive (AR and NAR) model assumes that the output of the model approximately follows a Gaussian distribution. This assumption is based on the idea that the current value of the time series is a linear combination of past values and some random noise, which is typically assumed to be normally distributed. The assumption of Gaussianity in the AR model allows for efficient parameter estimation using maximum likelihood estimation, as it simplifies the mathematical calculations and inference procedures. Moreover, many statistical techniques, such as hypothesis testing and confidence interval estimation, rely on the assumption of Gaussianity.

**Justification of stationery assumption in footnote 1.** We need to define traditional GC based on the assumption of stationarity before we can continue discussing the possibility of DWGC and DWGC-CI. From a practical perspective, it is more appropriate to understand this assumption as a very common and necessary preprocessing (smoothing) step, rather than simply referring to it as a stationarity assumption.

