# OpenReview forum: "Dynamic Window-level Granger Causality of Multi-channel Time Series"
_TMLR — Rejected by TMLR_

### Review · Reviewer_Cb6e · 2023-09-12

**Summary Of Contributions:**

This paper proposes a dynamic window Granger causality (DWGC) detection approach and explains why it can be improved by using a "causality index" (CI) matrix resulting in a DWGC-CI method. Theory is provided that shows that DWGC has a granularity-efficacy tradeoff, and that DWGC-CI has higher recall than DWGC. Experimental results on synthetic and real data show the effectiveness of the proposed DWGC-CI method.

**Audience:**

Yes

**Claims And Evidence:**

No

**Requested Changes:**

Please address the weaknesses that I brought up above. Overall, I think that the paper really needs to better compare the proposed method DWGC-CI with more baselines (even if it means using a dataset/setup that helps makes the comparison possible and that might not be one of the datasets currently considered) and provide much more discussion on why the assumptions made by the model make sense in practice (also compared against other methods). I would imagine that there is no single best model at present and I think that is okay, but providing this sort of context is very important and when and why people should use the proposed method vs a baseline method instead. Separately, the paper is littered in typos and is written in a manner right now that is simply not very clear. Please read over and polish the draft.

**Strengths And Weaknesses:**

Strengths:
- The problem setup is general and could be broadly applicable.
- The proposed method appears promising in theory and in experimental results.

Weaknesses:
- Very few experimental baselines are tested. I would have liked to see a more serious effort at benchmarking against more baselines, even if they are a bit more computationally expensive.
- I found the paper's exposition currently to be very confusing to follow and it doesn't help that there are way too many typos. Please proofread the paper carefully. To make matters worse, the citations right now really need to be fixed. If you do not mean to use an inline citation, then use \citep{}, whereas if you intend on using an inline citation, use \citet{}. Right now the citations are done in a manner that makes reading the paper to be difficult. For example, even the very first paragraph of the paper (in the introduction) has problems. It should read "Time series data, characterized by a pre-defined temporal or sequential order (Hamilton, 1994), is extensively employed in a diverse range of real-world applications, encompassing signal processing (Scharf, 1991), economics (Granger & Newbold, 2014), and climatology (Mudelsee, 2013), among others."
- More discussion of how realistic Assumption 1 is in practice would be helpful. To what extent does it apply to the real datasets considered?
- More discussion on the stationarity assumption in footnote 1 would also be helpful as well as it pertains to the datasets considered. Do any of the recently proposed methods mentioned in Section 2 not require this stationarity assumption?

---

> ### Author Response · Authors · 2023-10-04
> **Thank you for your thoughtful comment! Here is the response (1)**
>
> **C1**: Very few experimental baselines are tested. I would have liked to see a more serious effort at benchmarking against more baselines, even if they are a bit more computationally expensive.
>
> **A1**: We really appreciate your comments. As previously elucidated in the preliminary discussion, the integration of window-level and non-linear approaches represents a novel avenue of research, which consequently leads to a scarcity of well-established baselines. However, during the process of writing and revising the paper, some new work has indeed emerged that can serve as a baseline. As mentioned in our responses to reviewers 1 & 2, we have incorporated \[1\]\[2\]\[3\] into the baseline and compared the results and all the detailed information will be updated in the latest version. All these baselines on the real-world datasets have been tested and our DWGC-CI outperforms them significantly. We will kindly refer readers to our new version and Appendix which will be released soon.
>
> \[1\] Bodik, J., Paluš, M., & Pawlas, Z. (2021). Causality in extremes of time series. arXiv preprint arXiv:2112.10858.
>
> \[2\] Yang, W., Zhang, K., & Hoi, S. C. (2022). Causality-based multivariate time series anomaly detection. arXiv preprint arXiv:2206.15033.
>
> \[3\] Mastakouri, A. A., Schölkopf, B., & Janzing, D. (2021, July). Necessary and sufficient conditions for causal feature selection in time series with latent common causes. In International Conference on Machine Learning (pp. 7502-7511). PMLR.
>
> **C2**: I found the paper's exposition to be very confusing to follow and it doesn't help that there are way too many typos. Please proofread the paper carefully. To make matters worse, the citations right now really need to be fixed. If you do not mean to use an inline citation, then use \\citep{}, whereas if you intend on using an inline citation, use \\citet{}. Right now the citations are done in a manner that makes reading the paper to be difficult. For example, even the very first paragraph of the paper (in the introduction) has problems. It should read "Time series data, characterized by a pre-defined temporal or sequential order (Hamilton, 1994), is extensively employed in a diverse range of real-world applications, encompassing signal processing (Scharf, 1991), economics (Granger & Newbold, 2014), and climatology (Mudelsee, 2013), among others."
>
> **A2**: Thank you for your comment and we apologize for the typos. We have thoroughly revised the manuscript and will update it shortly.
>
> **C3**: More discussion of how realistic Assumption 1 is in practice would be helpful. To what extent does it apply to the real datasets considered?
>
> **A3**: Thanks for your detailed comment. We agree that the justification of key assumption is important, and can emphasize the generality of our theorems. We take the AR case for instance.
>
> In the context of time series modeling, the AutoRegressive (AR) model assumes that the output of the model approximately follows a Gaussian distribution. This assumption is based on the idea that the current value of the time series is a linear combination of past values and some random noise, which is typically assumed to be normally distributed. The assumption of Gaussianity in the AR model allows for efficient parameter estimation using maximum likelihood estimation, as it simplifies the mathematical calculations and inference procedures. Moreover, many statistical techniques, such as hypothesis testing and confidence interval estimation, rely on the assumption of Gaussianity.
>
> In contrast, the concept of "out-of-distribution" refers to data points that deviate significantly from the distribution on which the AR model has been trained. It represents cases where the observed data exhibits behaviors or patterns that are not well captured by the trained AR model. In other words, the model may struggle to make accurate predictions or provide a reliable estimate for data points that are outside the range of the learned distribution. Although the assumption of Gaussianity in the AR model relates to the specific distribution of the output, the concept of out-of-distribution focuses on the model's general ability to handle data that significantly differs from its trained distribution. Therefore, while the assumption of Gaussianity in the AR model allows for certain mathematical conveniences, the concept of out-of-distribution highlights the limitations of the model when facing data that it has not been adequately trained on.
>
> To summarize, the AR model assumes approximate Gaussianity in its output, facilitating parameter estimation and statistical inference. However, the concept of out-of-distribution considers the model's capability to handle data points that deviate significantly from its trained distribution and reflects its limitations in capturing extreme or uncommon patterns in the data. It can be considered as future work.

---

> > ### Author Response · Authors · 2023-10-04
> > **Thank you for your thoughtful comment! Here is the response (2)**
> >
> > **C4**: More discussion on the stationarity assumption in footnote 1 would also be helpful as well as it pertains to the datasets considered. Do any of the recently proposed methods mentioned in Section 2 not require this stationarity assumption?
> >
> > **A4**: Thank you for your question! In fact, the assumption of stability has always been necessary. We placed it in the footnote rather than explicitly in the Theorem section because this assumption is always implicit in any scientific research on Granger causality. We need to define traditional Granger causality based on the assumption of stationarity before we can continue discussing the possibility of DWGC and DWGC-CI. From a practical perspective, it is more appropriate to understand this assumption as a very common and necessary preprocessing (smoothing) step, rather than simply referring to it as a stationarity assumption. In conclusion, thank you for your careful reading and valuable thoughts.

---

### Review · Reviewer_L8bC · 2023-09-15

**Summary Of Contributions:**

The authors propose a procedure to perform dynamic Granger-causality analysis of multivariate time series in a non-stationary and nonlinear setting. Their algorithm is based on a window-level estimation of a nonlinear forecasting model and a F-test for pairwise Granger-causal relationships. The local fluctuations due to the sliding window approach are reduced by estimating a causality index matrix, which weights the observations in the F-test statistic.

The method, called DWGS-CI (dynamic window-level Granger causality with causality index), is tested on 2 synthetic and 2 real data sets, and shown to perform better than a variant of the method without the causality index matrix correction, DWGS, and a baseline method using a linear time series model and a rolling window.

The authors also show that under some assumptions on the forecasting model, the size of the rolling window is positively correlated with the recall rate of the GC analysis, and recall is higher for DWGS-CI than DWGS.

**Audience:**

Yes

**Broader Impact Concerns:**

No concern.

**Claims And Evidence:**

No

**Requested Changes:**

1. I think the experimental section needs to be strengthened. In particular, the synthetic data part should contain a multivariate setting with more than 2 dimensions. The possibility to detect spurious causalities in this case should also be tested.

2. To me, the method by rolling window should also be compared to methods that incorporate change-point detection such as [2], unless there is a good reason not to do so here. Also, as an ablation study, since the authors claim that their method is insensitive to the prediction model, it would be valuable to test what happens when a mis-specified VAR model is used instead of the NAR in the DWGS-CI algorithm. Besides, which nonlinear function is used in the NAR in Section 5?

3. The methodology for optimising the causality index matrix, which is, to my understanding, a new method, should be better described. In particular, what are the constraints on the entries or the norm of this matrix? Besides, the optimiser used in the experimental part should be specified in the appropriate section.

4. The authors should correct the general writing and formatting of the paper. There are some reference problems in the appendix (e.g., Theorem 3 which is Theorem 2 in the main paper if I am correct). Tables and Figures also need reformatting.

**Strengths And Weaknesses:**

Strengths:
- The task considered in this work is well introduced and the approach by rolling window to estimate dynamic Granger-causal relations is clearly described.
- The authors propose an interesting strategy based on a causality index matrix for mitigating the fluctuations introduced by the rolling window.
- The method is able to estimate different Granger-causal (GC) relationships in synthetic time series with a change point and recovers well-identified causalities existing in real data set.

Weaknesses:
- The claim of the paper that such method can perform dynamic GC analysis in multivariate time series is not sufficiently demonstrated. The empirical evidence is not sufficient and lacks comparison to other approaches, e.g., procedures that first estimate change-points in the multivariate time series.
- Some aspects of the methodology are not justified and discussed enough. For instance, the sensitivity on the choice of the threshold of the F-statistic is empirically tested but there is no general principled method proposed, e.g., to select significant values of the F-statistic. More importantly, the authors perform pairwise tests in multivariate time series, a strategy that is known to potentially detect spurious causalities in the presence of confounders [1]. I think this point should be discussed in the paper. Besides, I would also expect a discussion on other strategies for regulating the local fluctuations due to the window slicing, e.g., model parameters smoothing. Could the optimisation of the causality index matrix be avoided?
- The formatting and general writing of the paper are not polished. The limitations of previous methods mentioned in the literature review (Section 2) are not clear to me. Besides, the paper contains a lot of typos and the tables and figures are not sufficiently described.

---

> ### Author Response · Authors · 2023-10-04
> **Thank you for your thoughtful comment! Here is the response (1)**
>
> **C1** The experimental section needs to be strengthened. In particular, the synthetic data part should contain a multivariate setting with more than 2 dimensions. The possibility of detecting spurious causalities in this case should also be tested.
>
> **A1**: Thank you for your valuable suggestion! Specifically, in the current version, we have supplemented our experiments. We have expanded the 2-dimensional experiments to 20 dimensions. Our approach consists of two steps: 1) performing a 2-dimensional operation on any two dimensions to examine the window-level causal relationships, and 2) for all elements where $Y_{i_1}(t_{i_1})<-Y_{j}(t_j) -> Y_{i_2}(t_{i_2})$ is detected, we predict $Y_{i_1}$ and $Y_{i_2}$ sequences again after removing the information of $Y_j$ to eliminate the potential influence of confounders on causal identification. With this basic operation, our DWGC-CI method significantly outperforms all baselines. We will provide detailed explanations in the latest version. The causal graph upon multi-dimensional time series is defined as
>
> $$
> T_{i}(t) = Re(\sqrt{T_{j,t-lag}^2-1}) + n(t), i,j \in \{1,2,...M\}
> $$
> Here $M$ is the total sum of the number of channels.
>
> *Please note that the multidimensional approaches discussed here are only provided as examples and are not exhaustive. There are other more complex and sophisticated methods available such as \[1\]. The application and comparison of these methods can be considered as future work. We showcase the AR scenario for instance, and the whole performance will be shown in the latest version.
>
> | k   | 10  | 20  | 30  | 100 |
> | --- | --- | --- | --- | --- |
> | Extreme causal \[1\] (Baseline) | 0.31 0.46 | 0.55 0.51 | 0.75 0.93 | 0.86 **0.98** |
> | Anomal causal \[2\] (Baseline) | 0.23 **0.53** | 0.45 0.50 | 0.54 **0.94** | 0.66 0.95 |
> | Rolling-window GC (Baseline) | 0.21 0.42 | 0.28 0.51 | 0.52 **0.94** | 0.74 **0.98** |
> | DWGC (ours) | 0.58 0.41 | 0.59 0.75 | 0.71 0.92 | 0.86 0.98 |
> | DWGC-CI (ours) | **0.83** 0.43 | **0.88** **0.78** | **0.83** 0.93 | **0.87** **0.98** |
>
> **C2:** To me, the method by rolling window should also be compared to methods that incorporate change-point detection such as \[2\], unless there is a good reason not to do so here. Also, as an ablation study, since the authors claim that their method is insensitive to the prediction model, it would be valuable to test what happens when a misspecified VAR model is used instead of the NAR in the DWGS-CI algorithm. Besides, which nonlinear function is used in the NAR in Section 5?
>
> **A2**: Thanks for your sincere comments. 1. Compared with change-point detection, window-level causality detection has the following advantages:
>
> - **a) Finer time scales:** Window-level causality detection can analyze and detect causal relationships on finer time scales, whereas change-point detection usually analyzes based on the entire time series and cannot provide the same fine temporal resolution.
> - **b)Localized analysis:** Window-level causality detection divides the time series into consecutive windows and performs causality analysis within each window. This localized analysis can more accurately capture local variations and causality in the time series without being affected by the entire time series.
> - **c) Better sensitivity and specificity:** Window-level causality detection can analyze the data within each window, targeting causality judgments to the characteristics of the data within a specific window. This targeted analysis improves the sensitivity and specificity of the detection algorithm and reduces false positives and misses. Most importantly,
> - **d) Adaptation to dynamic changes:** Window-level causality detection can flexibly adapt to dynamic changes in the time series. The window size can be adjusted as needed to accommodate potential causal changes in different time periods. This dynamic adaptability makes window-level causality detection more applicable and robust. In sum, window-level causality detection has more advantages than variable-point detection in terms of time scale, analysis accuracy, sensitivity, and adaptability. The experiment also demonstrates the superiority of DWGC-CI over traditional change-point detection (we showcase the AR scenario).
>
> | *k* | *10* | *20* | *30* | *100* |
> | --- | --- | --- | --- | --- |
> | *Change point detection (Baseline)* | *0.38 0.31* | *0.39 0.59* | ***0.83** 0.90* | *0.80 **0.99*** |
> | *DWGC (ours)* | *0.58 0.41* | *0.59 0.75* | *0.71 0.92* | *0.86 0.98* |
> | *DWGC-CI (ours)* | ***0.83** **0.43*** | ***0.88** **0.78*** | ***0.83** **0.93*** | ***0.87** 0.98* |

---

> ### Author Response · Authors · 2023-10-04
> **Thank you for your thoughtful comment! Here is the response (2)**
>
> 2.  For the ablation study, we replace the well-trained NAR model with a coarse NAR without full training. We found that when the fitting accuracy of the NAR model slightly decreases, the accuracy and recall rate of causal effect identification are not significantly affected. This is because we focus on observing the ratio of the numerator and denominator of the F-statistics, rather than the specific values themselves. In NAR, the nonlinear function we adopt is the tanh(\\cdot) function, as mentioned in the method section.
>
> **C3**:The methodology for optimizing the causality index matrix, which is, to my understanding, a new method, should be better described. In particular, what are the constraints on the entries or the norm of this matrix? Besides, the optimizer used in the experimental part should be specified in the appropriate section.
>
> **A3**: Thanks for the comment.
>
> - **CI matrix.** For the motivation and constraints of the CI test, we kindly refer the reviewer to the response **A2** of **Reviewer  i4XB**. Briefly speaking, the CI matrix is to disentangle the noise from the true causal effect.
>
> - **Optimizer.** Moreover, for the optimizer used in the experimental part, in order to hold the generality, we adopt the most traditional Adam operator with the adaptive learning rate initialized as $\eta = 0.05$. We will add the total experimental details in the current version.
>
>
> **C4**: The authors should correct the general writing and formatting of the paper. There are some reference problems in the appendix (e.g., Theorem 3 which is Theorem 2 in the main paper if I am correct). Tables and Figures also need reformatting.\*\*\*
>
> **A4**: We appreciate the careful reading by the reviewer! We sincerely appreciate your contribution to the article. We sincerely apologize for any reading difficulties caused by some typographical errors. This was entirely our mistake and has been rectified with improvements in the latest version.
>
> \[1\] Mastakouri, A. A., Schölkopf, B., & Janzing, D. (2021, July). Necessary and sufficient conditions for causal feature selection in time series with latent common causes. In International Conference on Machine Learning (pp. 7502-7511). PMLR.

---

### Review · Reviewer_i4XB · 2023-09-20

**Summary Of Contributions:**

This work addresses the challenging task of detecting Granger causality (i.e., the predictive utility of one time-series channel to predict another) for time-series characterized by dynamic causalities.

Contrary to the classic Granger causality paradigm that assumes the causal effect between channels to be invariant over time, the authors aim at capturing window-level Granger causality, i.e., detecting causal relationships that evolve within sliding windows of a time-series.

The proposed methodology, named DWGC-CI, consists of:

- a windowing equivalent of the GC methodology (DWCG), where F-statistics within sliding windows of size $k$ are computed ---as per Equation (3)--- and,
- incorporating an indexing matrix $\Phi$ that scales down the fluctuations of the learned (within and across channels) forecasting models ---as per Equation (5).

The authors argue that the causality index (CI) matrix provides a dynamic, negative feedback regulation process to the DWGC computation, where $\Phi$ is learned to increase the recall rate of the proposed DWGC method.

Section 4 provides theoretical insights on DWGC ---when compared to classic GC--- and the impact of incorporating the CI matrix into the windowed GC framework.

In Section 5, empirical results are presented on synthetic and real-world datasets, showcasing  relative recall and accuracy improvements for the former, and providing qualitative arguments for the latter.

**Audience:**

Yes

**Broader Impact Concerns:**

This work does not discuss broader impact concerns.

**Claims And Evidence:**

No

**Requested Changes:**

1. Methodology:

   - In GC, the predictors are computed based on all the information available up to $t$: as indicated in Equation (1), one leverages data up to time instant $t$, i.e., $Y_{i,<t}$ indicates the time-series channel $i$ at and before time $t$. This approach makes sense in the GC framework, given that the underlying assumption is that the causality is invariant across time.
      - However, it is unclear what information is used to compute predictors $\hat{Y}_{i|q}$ and $\hat{Y}\_{i|j,q}$ in Equation (3): is information only within each window of the time-series used?
         - Using all previous information for each channel as in Equation (1) does not seem to correspond with the purpose of computing window-level causalities.
      - Clarifications and details on this matter are deemed necessary.

   - More detailed explanations and a rationale for computing the CI matrix $\Phi$ as in Equation (5) would be appreciated:

      - $q \in M$ in Equation (5) is said to relate to the set of sliding windows which detected causality ---contrary to the definition of time instants $q \in [t, t+k-1]$ in Equation (3)---: why is $\Phi$ computation limited to those windows? Isn't causality detection via F-statistics impacted by the $\Phi$ values themselves?
       - What is the expectation $\mathbb{E}_q$ inside $h$ referring to? Is this expectation over all sets $q \in M$ or all time instants $q \in [t, t+k-1]$?
       - Can the authors justify the choice of the argument of $h$ better? Why compute the difference between the squared differences of instantaneous errors and expectation over instantaneous errors?
       - Can the authors elaborate on why the need for the function $h$ to be mononotically decreasing? The authors propose a specific form in footnote 3, dependent on hyperparameter $\alpha$: can they elaborate on this parametric form and evaluate how sensitive is DWGC-CI to the choices of $h$?

2. Theory:

   - Theorem 1 assumes **channel level GC**, while DWGC aims at **window-level GC**: what is the significance of such Theorem?
      - To the best of my understanding, Theorem 1 clarifies that doing GC detection based on window-level information implies a performance loss under channel level GC, yet it does not provide insights on how accurate window-level GC detection is (i.e., the aim of this work).
      - I would appreciate it if the authors could clarify and expand on this important concern.

   - Theorem 2 argues for the existence of a causality matrix $\Phi$ under window-level GC that reduces the recall rate of DWGC-CI when compared to DWGC. However, details on how the definition of $L_{index}$ (e.g., the choice of h) and its minimization may allow us to achieve that goal are unclear.

   - I would encourage the authors to precisely lay down the assumptions needed for both Theorems to hold.

3. Experiments:

   - If my understanding is correct, Table 1 presents results for a single realization of each of the simulated AR and NAR time-series.
      - It would be interesting if the authors could replicate the experiments across different realizations and provide average and variability information of the results.

   - The simulated environments define "The time point of $m_i = 10, i = 1,2$" as the beginning of causality. Could the authors clarify how long is the causality considered to be active, and how it relates to windowing?
      - Are all windows of size $k$ that contain the time-instant where $m_i=10$ considered windows with GC?

   - The authors argue that DWGC-CI is "better than the baseline Ming-Hsien & Chih-She (2015), especially at k = 20, 100".
      - Can the authors elaborate on the dependency of the selected window size and windowed GC detection performance?
      - Why is performance not consistently better/worse with increasing/decreasing window sizes?

   - How long are the simulated time-series and how does it relate to the evaluated window-sizes?

   - The significance of the results could be strengthen with more simulations run to provide more clear and thorough evidence supporting the improved dynamic windowed GC detection claims.
      - For instance, given that the authors have simulated a linear scenario (AR), why not compare their approach to other, linear window-level causality baselines cited in the introduction?

4. Other suggested edits:

   - References to Figures, Equations and Assumptions are incorrectly typeset, as they appear as "Fig X equation X", "Eqn. equation X", "Assumption. equation X"

   - In text between Equations (1) and (2), it reads "check the difference between $\hat{Y}\_{i|j,t}$ and $\hat{Y}\_{i,t}$, i.e.,the forecasting result of the channel $j$ with and without channel $i$":
      - shouldn't it be "check the difference between $\hat{Y}\_{i|j,t}$ and $\hat{Y}\_{i,t}$, i.e.,the forecasting result of the channel $i$ with and without channel $j$"?

   - In the last paragraph of Section 2, the second to last sentence reads "are computational to verify", which seems to lack a clarifying word (e.g,. computationally hard?).

   - Before Equation 5, "We establish an minimization goal to learn" -> "We establish a minimization goal to learn"

   - Appendix Section D is entitled "Proof of Theorem 3", should be "Proof of Theorem 2"

**Strengths And Weaknesses:**

The main strengths of this work are:

- The significance of the studied problem: detecting window-level Granger causality is an important and relevant challenge.

- The windowing-based GC approach, and the corresponding per-window F-statistic computation, is sound.

- Results in synthetic experiments showcase a relative improvement of DWGC-CI over DWGC, and the selected alternative baseline.

There are several weaknesses of this work:

1. Lack of clarity and open questions on the proposed methodology (see requested clarifications in section below):

   - What time-series information is used to compute DWCG predictors $\hat{Y}\_{i|q}$ and $\hat{Y}\_{i|j,q}$ in Equation (3)? Are there as many predictors as windows within a time-series or is there a single predictor for each time-series channel?
   - Computation of Equation (5) is unclear and its reasonale demands further justification.
   - Is global convergence of the objectives in Algorithm 1 (NAR forecast loss and $L\_{index}$ minimization) simultaneously possible?

2. The theoretical analysis does not list all its assumptions, and the significance of the presented Theorems is unclear for window-level causality detection:

   - Assumption 1 relates to the unbiased nature of the predictors, while the proof of theorem 1 demands further assumptions (constant predictor variance, gaussian time-series innovations) that have not been clearly stated and argued for in the main manuscript.
   - Theorem 1, presented as a result on the positive correlation between recall rate and window length of DWGC, assumes **channel level GC**, while DWGC aims at **window-level GC**: the significance of such Theorem in the context of the goal of this work is unclear.
   - Assumptions for Theorem 2 (e.g., Gaussian weighted predictors) are not clearly laid out in the main manuscript.
   - Theorem 2 argues that causality matrix $\phi$ exists under window-level GC, yet it does not provide any insight on how the selected loss function $L_{index}$ minimization helps achieve such goal.

3. Empirical details and evaluation:

   - The proposed methodology requires learning 2 extra hyperparameters ($\epsilon$ and $\alpha$, for fixed $h$), besides the selection of time-series window size $k$ and form of $h$.
   - Synthetic results (with ground truth access) are provided for only 2 simulated realizations ---which showcase a promising, yet inconclusive, performance improvement. The evaluation with real datasets seems subject to qualitative expert knowledge ---due to a lack of ground truth.

---

> ### Author Response · Authors · 2023-10-04
> **Thank you for your thoughtful comment! Here is the response (1)**
>
> **C1**: In GC, the predictors are based on all the information available up to t as indicated in Equation (1), one leverages data up to time instant, i.e., indicates the time-series channel at and before the time. This approach makes sense in the GC framework, given that the underlying assumption is that the causality is invariant across time. But what about DWGC?
>
> **A1**: DWGC makes a compromise on time-invariance and varying and takes advantage of the information of the channel up to not only the information of the current window but also the information of the channel up to time point $t$. The task of DWGC is to accurately determine whether channel $i$ has received a causal effect within the window $t\sim t+k-1$. We will make it clear in the revision.
>
> In detail, compared with Equations (1) and (3) in the main text, the only difference is located in the selection of the window size instead of what information the predictor uses. Consider this scenario: if the information from channel $i$ cannot fit well with the information from window $t\sim t+k-1$, but adding the information from channel $j$ makes it possible—this can be roughly considered as the possibility that channel $i$ is causally influenced within the window $t\sim t+k-1$. Conversely, if we discard the information outside the window $t\sim t+k-1$ and do not include it as inputs, we cannot reach the above conclusion based on extremely limited data information. To be honest, in order to avoid misunderstandings and ambiguities, the indicator expressed by Equation (3) in the text better refers to the causal relationship between $Y_j(0-t)$ and $Y_i(t-t+k-1)$. We will make this revision in the current version. Noteworthy, this statement does not affect the various conclusions of the article, as it is not mandatory for us to precisely locate the starting and ending points of each causal relationship (which is quite difficult to accomplish in a time series). Thanks for this valuable question!
>
> **C2**: More detailed explanations and a rationale for computing the CI matrix as in Equation (5) would be appreciated:
>
> **A2**: Thanks for the comment. We will give two more aspects of explanation for Equation (5):
>
> - **The reason for not updating CI on the entire area.** The key to this issue lies in the motivation behind introducing the CI matrix: we want to increase the credibility of areas where there may be potential causal relationships with significant fluctuations, by excluding the influence of noise. The specific approach is to weaken the impact of excessive outliers in these "suspicious areas" (as they are likely noise). This compels us to focus on updating the coefficients of the CI matrix in the "suspected" regions of causal relationships, rather than considering the entire area. It should be noted that this weighted filtering pattern is dynamic because the extracted set of causal relationship windows may vary in each round. Our experiment results could further demonstrate this motivation since the performance is weaker if we conduct an update for all $\Phi$ on all windows. We have added these comparisons in the current version.
>
> Notice that here $q$ denotes the interval and $q \in M$ is the set of sliding windows that detected causality.
>
> - **The justification of loss function.** Note that due to the unstable and easily disrupted nature of DWGC under window settings, when this indicator significantly increases, there are two factors at play: a) the genuine causal effect (observing when to channel $j$ influences $i$), and b) the local window noise (observing outliers of the instantaneous errors in the autoregressive model of $i$). The calculation of differences between instantaneous errors and the expectation over instantaneous errors aims to disentangle these two factors. Specifically, in the windows where potential causal relationships exist, we actively detect significant outliers in this set of instantaneous errors. These outliers are more likely to be influenced by real-time noise compared to other points. Through a negatively correlated (monotonically decreasing) function h, we apply negative feedback adjustment to the weighted values (CI matrix) corresponding to these outliers. This aids in reducing the impact of these noisy points on the final result's weight (since the absolute prediction error is often proportional to the scale of the data points themselves).
>
> Moreover, the monotone-deceasing guarantee of $h$ has been illustrated above, and its specific form $\alpha-tanh(\cdot)$ is inspired by the active function in the neural network in order to highlight the informative points. The parameter $\alpha$ is initialized as $1.2$ in order to make sure the expectation is nearly $1$ (the choice is not unique). In our experiments, we also showcase the sensitivity analysis of the hyper-parameter, which will be illustrated in the revised version. In sum, thank you very much for your attention to detail, as it can stimulate further discussion.

---

> ### Author Response · Authors · 2023-10-04
> **Thank you for your thoughtful comment! Here is the response (2)**
>
> **C3**: Theorem 1 assumes channel-level GC, while DWGC aims at window-level GC: what is the significance of such Theorem's details on how the definition of (e.g., the choice of h) and its minimization may allow us to achieve that goal are unclear. I would encourage the authors to precisely lay down the assumptions needed for both Theorems to hold.
>
> **A3**: Thanks for the comment and we would like to clarify accordingly:
>
> - **Theorem 1.** Thank you for your thoughtful comments. In short, the significance of Theorem 1 is that it does not directly indicate the core contribution of the article, but rather highlights the core motivation behind the research. In fact, we adopt a progressive logical organization for our theoretical contributions: 1) We present an important and interesting **'Negative Result'** regarding the performance of traditional GC methods at the WINDOW-LEVEL: their performance gradually worsens as the window size decreases, which aligns with our intuition that "window-level instability" is prone to occur; 2) This negative result serves as a solid foundation for the innovative design of DWGC-CI. We need to introduce a new force, namely the dynamically adjustable CI matrix, to counteract the diminishing power of the GC model as the window size decreases. We delightedly believe that a **'Negative Result'** with good insight is equally crucial as it represents a perspective of thinking about a problem from another angle.
>
> - **Theorem 2.** We gently invite the reviewers to refer to the appendix section. In Theorem 3 and Lemma 2, we propose a more complex form of the CI matrix, which provides a sufficient condition for the superiority of DWGC-CI over DWGC. In fact, as long as we actively ensure that $L_{index}$ can converge near this solution, we can guarantee the superiority of DWGC-CI. At the same time, we honestly acknowledge that there is currently no conclusion on how to rigorously analyze the convergence conditions of CI under $L\_{index}$ due to the lack of clear mathematical tools. This presents an interesting theoretical direction for future work. However, it is gratifying that in experiments, DWGC-CI has demonstrated superior performance, confirming that good CI is often easy to obtain.
>
> - **Assumption.** Furthermore, the assumption for these two theorems to hold is summarized by Assumption 1. More strictly, in Theorem 1, the outputs satisfy an unbiased and stable Gaussian distribution with the same magnitude of fluctuations at each data point, i.e., the Gaussian distributions of these outputs share a uniform noise variance when doing autoregression and regression of reciprocity, respectively; in Theorem 2, we further assume that if the data are weighted using the CI-matrix, the output will also be a Gaussian distribution with the same scale of deflation. These assumptions are realistic, and we have put all the rigorous assumptions in the text in the latest version. Thank you for your excellent suggestions. We deeply apologize for the typographical error of including an extra "equation" within it.
>
>
> **C4**: It would be interesting if they could replicate the experiments across different realizations and provide average and variability information on the results.
>
> **A4**: Thank you for your suggestion, as it helps us demonstrate the robustness of our method. We fully adhere to your advice and have included the additional experiments as suggested：(The above is the recall rate and the below is the accuracy rate, we show the AR case for instance. The total performance will be shown in the newest version.)
>
> | k   | 10  | 20  | 30  | 100 |
> | --- | --- | --- | --- | --- |
> | Extreme causal \[1\] (Baseline) | 0.30(0.018) 0.47(0.013) | 0.56(0.013) 0.52(0.016) | 0.76(0.009) 0.94(0.010) | 0.87(0.005) 0.99(0.003) |
> | Anomal causal \[2\] (Baseline) | 0.22(0.014) **0.54(0.013)** | 0.43(0.018) 0.52(0.015) | 0.53(0.009) **0.95(0.010)** | 0.65(0.004) 0.96(0.004) |
> | Rolling-window GC (Baseline) | 0.20(0.011) 0.43(0.013) | 0.29(0.010) 0.52(0.011) | 0.53(0.009) **0.95(0.010)** | 0.75(0.002) 0.99(0.002) |
> | DWGC (ours) | 0.59(0.014) 0.42(0.019) | 0.60(0.009) 0.76(0.010) | 0.72(0.011) 0.93(0.013) | 0.87(0.007) 0.99(0.008) |
> | DWGC-CI (ours) | **0.84(0.009)** 0.44(0.009) | **0.89(0.007)** **0.79(0.009)** | **0.84(0.005)** 0.94(0.005) | **0.88(0.005)** **0.99(0.001)** |
>
> It can be seen that our DWGC-CI method is more robust than the baseline, and the results gradually stabilize as the window length increases.
>
> **C5**: Are all windows of size that contain the time-instant considered windows with GC?
>
> **A5**: Yes. Moreover, we use $lag$ to control the causality, which is understood as $Y_{i}(t) \rightarrow Y_{j}(t+lag)$ and considered as an 'active causality'. In our experiments, we separate the whole channel with different window sizes $d=10,20,30,100$. within each sub-window, if any $Y_i(t)$ is pointed by $Y_j(t-lag)$ and the corresponding $m_i$ contains $10$, then it is identified as causality.

---

> > ### Author Response · Authors · 2023-10-04
> > **Thank you for your thoughtful comment! Here is the response (3)**
> >
> > **C6**: Can the authors elaborate on the dependency of the selected window size and windowed GC detection performance? Why is performance not consistently better/worse with increasing/decreasing window sizes?
> >
> > **A6**: We can observe that:
> >
> > - With the increase in window length, both the baseline method and our method (Rolling window method, DWGC, DWGC-CI) show an overall improvement in accuracy and recall. This is because as the window length increases, the situation degrades into a traditional GC, and the model's performance becomes more stable.
> > - From a vertical comparison perspective, we found that our method DWGC, especially DWGC-CI, significantly outperforms the baseline method (Rolling-window method). Moreover, this superiority becomes more significant when the window size is smaller, as our CI matrix is specifically designed to address the instability of local windows.
> > - Furthermore, the out-performance is more significant in NAR than in AR, the potential reason is that in the NAR scenario, it is more challenging to decouple local noise and causal effects, and the performance becomes simultaneously worse. Therefore, the baseline method, which does not focus on distinguishing noise, exhibits a larger performance gap in the NAR scenario.
> > - It is worth noting that this result is not consistent in all cases, as the model performance of the baseline method is unstable and biased on local windows. When we supplement variation analysis, we find that the results are basically consistent.
> >
> > **C7**: How long are the simulated time series and how does it relate to the evaluated window sizes?
> >
> > **A7**: We replicate the experiment and choose the whole length as $L=10000$. It is separated into the $\frac{L}{k}$ windows with window size $k$ (The first window is omitted).
> >
> > **C8**: The significance of the results could be strengthened with more simulations run to provide more clear and thorough evidence supporting the improved dynamic windowed GC detection claims. For instance, given that the authors have simulated a linear scenario (AR), why not compare their approach to other, linear window-level causality baselines cited in the introduction?
> >
> > **A8**: As previously elucidated in the preliminary discussion, the integration of window-level and non-linear approaches represents a novel avenue of research, which consequently leads to a scarcity of well-established baselines. However, in order the emphasize the validity of our method, we add baselines\[1,2\] during the revision of our work, as much as possible, to the best of our knowledge.
> >
> > We showcase the performances of these methods as in the table above, for additional performance of these methods on the real-world datasets, we politely refer readers to the newest version of our paper, which will be updated shortly.
> >
> > **A9-typos**: Thanks for pointing out the typos and we have carefully reviewed and modified the writing in the latest version. We sincerely appreciate your contribution to the article.
> >
> > \[1\] Bodik, J., Paluš, M., & Pawlas, Z. (2021). Causality in extremes of time series. arXiv preprint arXiv:2112.10858.
> >
> > \[2\] Yang, W., Zhang, K., & Hoi, S. C. (2022). Causality-based multivariate time series anomaly detection. arXiv preprint arXiv:2206.15033.

---

### Author Response · Authors · 2023-10-04
**Thank you for your valuable suggestions!**

We would like to thank all the reviewers for their valuable insights and suggestions which have helped thoughtfully improve the manuscript. We are revising our paper accordingly and we reply to the comments of every reviewer individually. We will update the draft shortly.

---

### Decision · Action_Editor_34ay · 2023-10-27

**Recommendation:** Reject

**Comment:**

The authors propose a novel approach for dynamic Granger-causality analysis in nonstationary and nonlinear settings. Their method, DWGS-CI, combines a window-level estimation of a nonlinear forecasting model with F-tests for pairwise Granger-causal relationships. It outperforms the previous methods in tests on synthetic and real datasets, demonstrating the value of incorporating a causality index in the analysis.

All reviewers appreciate the authors' efforts in responding to their comments and making revisions to the manuscript. However, they have raised several key concerns about the clarity and theoretical foundation of the work. After author rebuttal, the reviewers still have the following concerns during reviewer discussion and the majority of them do not recommend acceptance of the current version.

Firstly, the main claim, which involves the incorporation of the causality index to stabilize the window-level approach, remains inadequately justified. Both reviewers L8bC and i4XB found this aspect lacking in clear theoretical support. They emphasize that the significance and scope of the theoretical results need to be more clearly articulated. Additionally, Reviewer L8bC expressed concerns about the marginal improvement in the revised manuscript and suggested that the added clarifications should be integrated into the text, as the content currently lacks good organization and description, impacting the overall readability. Reviewer  i4XB highlighted the importance of a more comprehensive discussion and evaluation of the practical aspects of the computation method, which currently appears somewhat ad-hoc.

Moreover, the theoretical foundation of the work was questioned by Reviewer i4XB. In Theorem 1, the performance of DWGC affecting by the window size, is not adequately informative and fails to justify various aspects of the proposed framework, such as the choice of windowing or the specific formulation of the causality index. Additionally, assumptions about Gaussianity and constant variance underlying the presented theorems were too restrictive from a practical perspective.

In summary, the current manuscript needs a major revision. I have to recommend rejection because the reviewing system does not allow a second round of major revision, but I encourage the authors resubmit the paper to TMLR after another round of revision.

**Audience:**

Yes

**Claims And Evidence:**

No.

1. There are several key concerns about the clarity and theoretical foundation of the work.
2. The main claim, which involves the incorporation of the causality index to stabilize the window-level approach, remains inadequately justified.

**Resubmission Of Major Revision:**

The authors may consider submitting a major revision at a later time.